# Lowland plant arrival in alpine ecosystems facilitates a decrease in soil carbon content under experimental climate warming

Tom WN Walker[1]*, Konstantin Gavazov[2,3,4], Thomas Guillaume[3,4,5], Thibault Lambert[6], Pierre Mariotte[3,4,7], Devin Routh[1], Constant Signarbieux[3,4], Sebastián Block[1,8], Tamara Münkemüller[9], Hanna Nomoto[1], Thomas W Crowther[1], Andreas Richter[10,11], Alexandre Buttler[3,4], Jake M Alexander[1]

[1]Institute of Integrative Biology, ETH Zürich, Zürich, Switzerland; [2]Climate Impacts Research Centre, Department of Ecology and Environmental Sciences, Umeå Universitet, Abisko, Sweden; [3]Swiss Federal Institute for Forest, Snow and Landscape Research (WSL), Lausanne, Switzerland; [4]École Polytechnique Fédérale de Lausanne EPFL, School of Architecture, Civil and Environmental Engineering ENAC, Laboratory of Ecological Systems ECOS and Plant Ecology Research Laboratory PERL, Lausanne, Switzerland; [5]Field-Crop Systems & Plant Nutrition, Nyon, Switzerland; [6]Faculty of Geosciences & the Environment, Université de Lausanne, Lausanne, Switzerland; [7]Grazing Systems, Agroscope, Posieux, Switzerland; [8]Department of Ecology & Evolutionary Biology, Princeton University, Princeton, United States; [9]Uni. Grenoble Alpes, CNRS, Uni. Savoie Mont Blanc, LECA, Laboratoire d'Ecologie Alpine, Grenoble, France; [10]Centre of Microbiology & Environmental Systems, Division of Terrestrial Ecosystem Research, University of Vienna, Vienna, Austria; [11]International Institute for Applied Systems Analysis, Ecosystem Services and Management Program, Laxenburg, Austria

*For correspondence: thomas.walker@unine.ch

**Competing interest:** The authors declare that no competing interests exist.

**Abstract** Climate warming is releasing carbon from soils around the world, constituting a positive climate feedback. Warming is also causing species to expand their ranges into new ecosystems. Yet, in most ecosystems, whether range expanding species will amplify or buffer expected soil carbon loss is unknown. Here, we used two whole-community transplant experiments and a follow-up glasshouse experiment to determine whether the establishment of herbaceous lowland plants in alpine ecosystems influences soil carbon content under warming. We found that warming (transplantation to low elevation) led to a negligible decrease in alpine soil carbon content, but its effects became significant and 52% ± 31% (mean ± 95% confidence intervals) larger after lowland plants were introduced at low density into the ecosystem. We present evidence that decreases in soil carbon content likely occurred via lowland plants increasing rates of root exudation, soil microbial respiration, and $CO_2$ release under warming. Our findings suggest that warming-induced range expansions of herbaceous plants have the potential to alter climate feedbacks from this system, and that plant range expansions among herbaceous communities may be an overlooked mediator of warming effects on carbon dynamics.

## Editor's evaluation

The authors transplanted alpine turfs from their cold environment to a lowland warm environment. They found that when lowland plants were inserted into these turfs under the thus simulated warming treatment they rapidly increased soil microbial decomposition of carbon stocks due to root exudates feeding the soil microbes. This finding is relevant because it suggests that global warming and shifts in plant species distributions may cause the release of soil-stored carbon into the atmosphere, thus further increasing warming.

## Introduction

Climate warming is expected to accelerate the release of soil carbon to the atmosphere (*Canadell et al., 2021*). At the same time, warming is driving uphill and poleward range expansions of native plant species around the globe (*Parmesan, 2006*; *Lenoir and Svenning, 2015*; *Pecl et al., 2017*; *Dolezal et al., 2016*; *Rumpf et al., 2018*; *Steinbauer et al., 2018*). Yet while warming and plant range expansions are now co-occurring in many terrestrial ecosystems, the impact of range expanding plants on soil carbon storage under warming remains poorly understood (*Wallingford et al., 2020*). This is important because plant species adapted to different temperatures can have contrasting effects on carbon cycle processes (*Walker et al., 2019*), raising the potential for migrating species to have qualitatively different impacts on carbon dynamics from resident plants. For example, studies of non-native species invasions have revealed that the arrival of novel species in a community can alter carbon cycling in recipient ecosystems (*Liao et al., 2008*; *Waller et al., 2020*), while woody plant expansions into treeless Arctic ecosystems can reverse the direction of carbon cycle feedbacks to climate warming (*Pearson et al., 2013*; *Cornelissen et al., 2007*). However, the majority of warming-induced range expansions do not involve non-native species or novel growth forms, but occur among native communities of similar species (e.g. *Steinbauer et al., 2018*). In particular, we know remarkably little about how migrations of herbaceous plants into predominantly herbaceous ecosystems will affect carbon cycling, and so the magnitude and direction of resulting climate feedbacks are unknown.

Alpine ecosystems occur in the uppermost elevations of the planet as cool islands within warmer landscapes that are bounded by steep gradients in temperature (*Körner, 2003*). They are experiencing both rapid temperature change (0.4–0.6°C per decade) (*Pepin, 2015*; *Vitasse et al., 2021*) and rapid species immigration (*Lenoir et al., 2008*), the latter of which involves an influx of herbaceous warm-adapted species from lowlands into herbaceous cold-adapted communities at higher elevations (*Dolezal et al., 2016*; *Rumpf et al., 2018*; *Steinbauer et al., 2018*). Coupled to this, alpine ecosystems are globally relevant reservoirs of soil carbon and hold more than 90% of their total ecosystem carbon in the soil (*Streit et al., 2014*; *Parker et al., 2015*; *Knowles et al., 2019*; *Hagedorn et al., 2019*; *Budge et al., 2011*). As such, alpine ecosystems are not only themselves important for climate feedbacks, but also sentinels for climate warming impacts on other terrestrial ecosystems. Here, we used alpine ecosystems as a model system to determine how the establishment of range expanding native plant species could affect soil carbon storage in warming ecosystems. We first used two whole-community transplant experiments in different alpine regions (western/central Swiss Alps) to establish the effects of warming plus lowland plant arrival on alpine soil carbon content. Second, we used these experiments plus glasshouse and laboratory studies to explore potential mechanisms through which lowland plants influence the alpine soil carbon cycle, testing the hypothesis that they act via changes to root exudation (*Hafner et al., 2012*; *Studer et al., 2014*).

## Results and discussion
### Lowland plants facilitate warming-induced soil carbon loss

We exploited two independent field-based transplant experiments in which alpine plant communities plus rooting zone soil were transplanted from high to low elevation to simulate a climate warming scenario, or back into a high-elevation site as a negative control (*Appendix 1—figure 1*; Materials and methods). A portion of the alpine turfs at low elevation were planted with local lowland plant species to simulate the arrival of warm-adapted lowland plants in the warmed ecosystem, with the remaining portion being subjected to a planting disturbance control. Elevation-based transplant experiments are powerful tools for assessing climate warming effects on ecosystems because they expose plots to a real-world future temperature regime with natural diurnal and seasonal cycles while also warming

**eLife digest** In a terrestrial ecosystem, the carbon cycle primarily represents the balance between plants consuming carbon dioxide from the atmosphere and soil microbes releasing carbon stored in the soil into the atmosphere (mostly as carbon dioxide). Given that carbon dioxide traps heat in the atmosphere, the balance of carbon inputs and outputs from an ecosystem can have important consequences for climate change.

Rising temperatures caused by climate warming have led plants from lowland ecosystems to migrate uphill and start growing in alpine ecosystems, where temperatures are lower and most carbon is stored in the soil. Soil microbes use carbon stored in the soil and exuded from plants to grow, and they release this carbon – in the form of carbon dioxide – into the atmosphere through respiration. Walker et al. wanted to know how the arrival of lowland plants in alpine ecosystems under climate warming would affect carbon stores in the soil.

To answer this question, Walker et al. simulated warmer temperatures by moving turfs (plants and soil) from alpine ecosystems to a warmer downhill site and planting lowland plants into the turfs. They compared the concentration of soil carbon in these turfs to that of soil in alpine turfs that had not been moved downhill and had no lowland plants. Their results showed that the warmed turfs containing lowland plants had a lower concentration of soil carbon. This suggests that climate warming will lead to more soil carbon being released into the atmosphere if lowland plants also migrate into alpine ecosystems.

Walker et al. also wanted to know the mechanism through which lowland plants were decreasing soil carbon concentration under warming. They find that lowland plants probably release more small molecules into the soil than alpine plants. Soil microbes use the carbon and nutrients in these molecules to break down more complex molecules in the soil, thereby releasing nutrients and carbon that can then be used in respiration. This finding suggests that soil microbes breakdown and respire native soil carbon faster in the presence of lowland plants, releasing more carbon dioxide into the atmosphere and reducing carbon stores in the soil.

Walker et al.'s results reveal a new mechanism through which uphill migration of lowland plants could increase the effects of climate change, in a feedback loop. Further research as to whether this mechanism occurs in different regions and ecosystems could help to quantify the magnitude of this feedback and allow scientists to make more accurate predictions about climate change.

both above- and belowground subsystems (*Hannah, 2022*; *Shaver et al., 2000*; *Yang et al., 2018*). This is especially true if they include rigorous disturbance controls (here, see Materials and methods) and are performed in multiple locations where the common change from high to low elevation is temperature (here, warming of 2.8°C in the central Alps and 5.3°C in the western Alps). While factors other than temperature can co-vary with elevation (*Körner, 2003*), such factors either do not vary consistently with elevation among experiments (e.g. precipitation, wind), are not expected to strongly influence plant performance (e.g. UV radiation) or in any case form part of a realistic climate warming scenario (e.g. growing-season length, snow cover) (*Körner, 2003*; *Yang et al., 2018*; *Nomoto and Alexander, 2021*). In short, the experiments used here examined how the arrival of warm-adapted lowland plants influences alpine ecosystems in a warmed climate matching lowland site conditions (i.e. turf transplantation to low elevation plus lowland plant addition) relative to warming-only (i.e. turf transplantation to low elevation) or control (i.e. turf transplantation within high elevation) scenarios.

We found that transplantation from high to low elevation (i.e. warming-only) caused a small decline in alpine soil carbon content, but in both regions this effect was stronger and only became statistically significant when lowland plants were also planted-in (*Figure 1*; *Supplementary file 1*; p < 0.001). Specifically, considering field data from both experiments together, the addition of lowland plants to transplanted alpine turfs increased warming-induced soil carbon loss from 4.1 ± 0.23 mg g$^{-1}$ per °C to 6.3 mg g$^{-1}$ per °C, which is a 52% ± 31% (mean ± 95% confidence intervals [CIs]) increase relative to warming alone (*Figure 1—figure supplement 1*; p = 0.001). We caution against equating changes to soil carbon content with changes to soil carbon stock in the absence of coupled measurements of soil bulk density (Materials and methods). Nevertheless, these findings show that once warm-adapted lowland plants establish in warming alpine communities, they facilitate warming effects on soil carbon

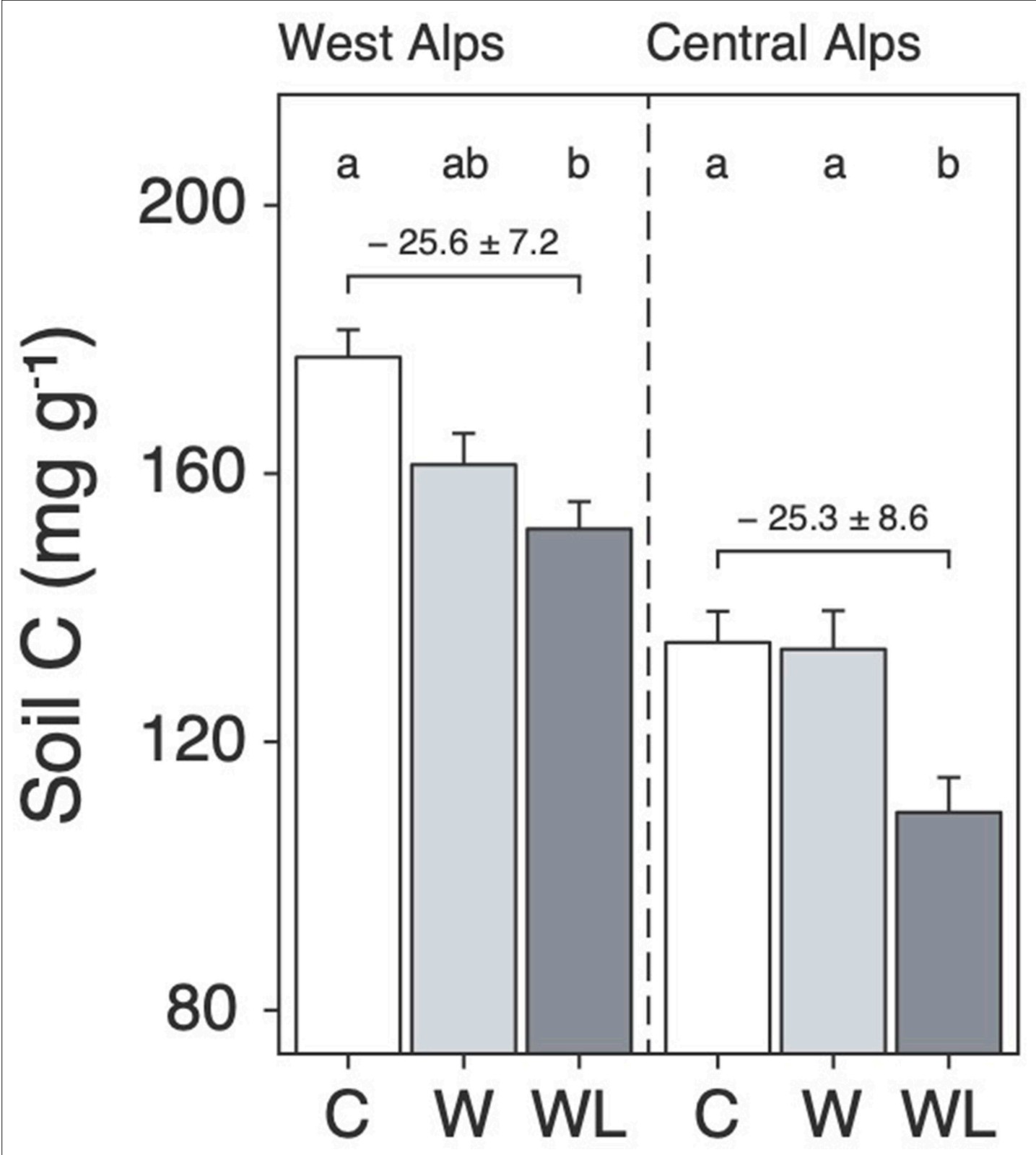

**Figure 1.** Warming and lowland plant effects on alpine soil carbon content in the field experiment. Mean ± standard error (SE) soil carbon content (mg C g$^{-1}$ dry mass; i.e. mass-based per-mil) in alpine turfs transplanted to low elevation (warming, W; light grey), transplanted plus planted with lowland plants (warming plus lowland plant arrival, WL; dark grey), or replanted at high elevation (control, C; white). Data are displayed for two experiments in the western (left) and central (right) Alps, with letters indicating treatment differences (linear mixed effects models; $N$ = 58).

*Figure 1 continued on next page*

*Figure 1 continued*

The online version of this article includes the following figure supplement(s) for figure 1:

**Figure supplement 1.** Soil temperature and lowland plant effects on alpine soil carbon content.

loss on a per gram basis. We thus suggest that uphill migrations of lowland plants, which are already globally common (*Rumpf et al., 2018*; *Lenoir et al., 2008*), may have a hitherto unrecognized bearing over alpine soil carbon storage.

## Lowland and alpine plants have distinct effects on soil microbial respiration

Soil carbon loss occurs due to a shifting balance between soil carbon inputs via net primary production and soil carbon outputs via soil microbial respiration (*Allison et al., 2010*; *Liang et al., 2017*). We therefore hypothesized that lowland plants impacted alpine soil carbon loss by either decreasing net primary production or increasing soil microbial respiration relative to alpine plants. To test this, we performed a glasshouse experiment in which we grew lowland plants (*Bromus erectus*, *Salvia pratensis*, *Medicago lupulina*) or alpine plants (*Phleum alpinum*, *Anthyllis vulneraria* subsp. *alpestris*, *Plantago alpina*) in alpine soil collected from the western Alps region. The glasshouse experiment was not a direct simulation of field conditions, in that plants from lowland and alpine communities were grown separately at a constant temperature and humidity (Materials and methods). Instead, the glasshouse experiment allowed us to isolate how plants adapted to lowland versus alpine climates differentially affect the alpine soil system.

We first compared lowland and alpine plant traits to test whether lowland plants decreased carbon inputs into alpine soil. We found that lowland plant treatments produced more aboveground (*Figure 2a*; *Supplementary file 2*; p = 0.028) and belowground (*Figure 2b*; *Supplementary file 2*; p < 0.001) biomass than alpine plants, while also containing a higher proportion of root tissue (root-to-shoot ratio: *Figure 2c*; *Supplementary file 2*; p = 0.005). While biomass is an imperfect proxy for net primary production, these findings point away from the hypothesis that lowland plants cause alpine soil carbon loss by reducing carbon inputs into soil. On the contrary, the higher biomass and proportionately more root tissue present in lowland plant treatments suggest that lowland plants may in fact increase soil carbon inputs. This occurred despite lowland plant treatments possessing similar specific leaf area (SLA: *Figure 2d*; *Supplementary file 2*; p = 0.861) to alpine plant treatments, as well as lower maximal photosynthetic capacity ($A_{max}$: *Figure 2e*; *Supplementary file 2*; p = 0.026) and stomatal conductance ($g_s$: *Figure 2f*; *Supplementary file 2*; p < 0.001). While the reason for decreased photosynthetic capacity and stomatal conductance in the lowland plant treatment is

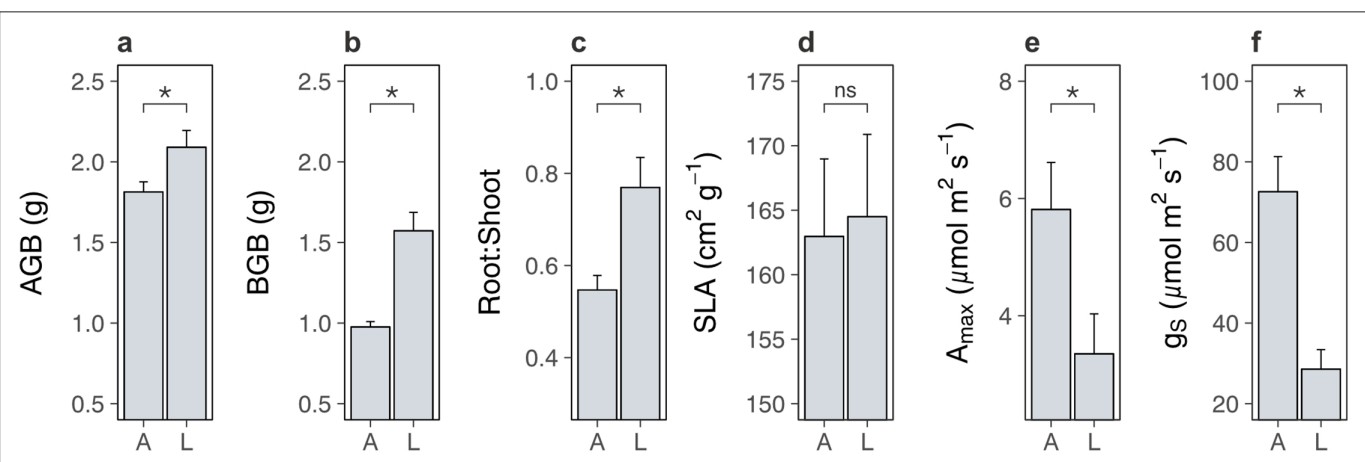

**Figure 2.** Differences between lowland and alpine plant traits in the glasshouse experiment. Mean ± standard error (SE) pot-level traits for alpine (A) and lowland (L) plant treatments in a glasshouse experiment. (**a**) Aboveground biomass (AGB; g; N = 19). (**b**) Belowground biomass (BGB; g; N = 10). (**c**) Root-to-shoot ratio (N = 19). Biomass-weighted (**d**) specific leaf area (SLA; cm$^2$ g$^{-1}$; N = 18), (**e**) maximum photosynthetic capacity ($A_{max}$; μmol m$^2$ s$^{-1}$; N = 19), and (**f**) maximum stomatal conductance ($g_s$; μmol m$^2$ s$^{-1}$; N = 20). Treatment effects (p < 0.05) are indicated by asterisks (LMEs; see *Supplementary file 2*).

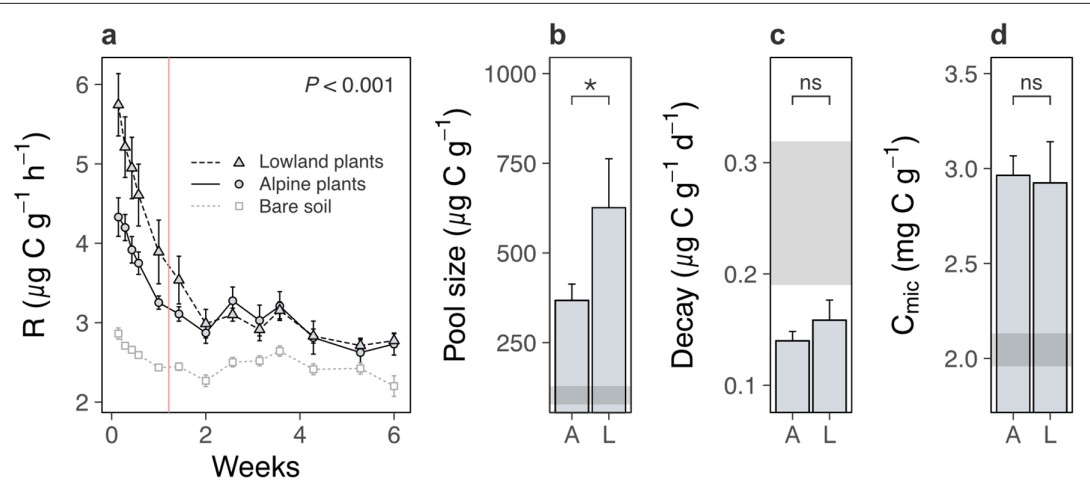

**Figure 3.** Lowland and alpine plant effects on alpine soil microbes in the glasshouse experiment. Effects of alpine (A) versus lowland (L) plant treatments on alpine soil microbes in a glasshouse experiment (mean ± standard error [SE]; grey points/ribbons: SEs for bare soil). (**a**) Total microbial respiration ($R$; µg C g$^{-1}$ dry mass h$^{-1}$) during follow-up incubations without plants (N = 260). (**b**) Size (µg C g$^{-1}$ dry mass) and (**c**) decay rate (µg C g$^{-1}$ dry mass d$^{-1}$) of a carbon pool identified using two-pool models (N = 16). (**d**) Microbial biomass carbon ($C_{mic}$; mg C g$^{-1}$ dry mass; N = 18). For (**a**), vertical line indicates time before which A–L comparison differed (LME, Tukey; see Main text). For (**b–d**), treatment effects (p < 0.05) are indicated by asterisks (LMEs; see *Supplementary file 2*).

unclear, the fact that both variables were lower suggests that photosynthetic capacity was limited by stomata pore size rather than by photosynthetic machinery (*Wong et al., 1979*). Indeed, lowland plants accumulated more biomass despite having a lower maximal photosynthetic capacity, ruling it out as a limiting factor of lowland plant net primary production.

We next examined the soil system under lowland versus alpine plants to determine whether lowland plants alternatively increased carbon outputs from alpine soil. We found that lowland plants accelerated soil microbial respiration relative to alpine plants (*Figure 3a*; treatment × time: likelihood ratio [LR] = 147.99, p < 0.001). Accelerated microbial activity was also reflected in spectral analyses of soil pore water, which showed that soil under lowland plants contained more dissolved organic matter (DOM) (*Figure 4a, b*; *Supplementary file 2*; absorbance [$a_{350}$]: p < 0.001, total fluorescence [$F_{tot}$]: p < 0.001) with a higher proportion of microbially derived products (*Figure 4c*; *Supplementary file 2*; fluorescence index [FI]; p = 0.002) than soil under alpine plants. Importantly, microbial respiration was enhanced despite no changes to microbial biomass between treatments (*Figure 3d*; *Supplementary file 2*; p = 0.693), showing that microbes respired intrinsically faster (i.e. per unit of biomass) in the presence of lowland plants. These findings support the hypothesis that lowland plants have the capacity to increase soil carbon outputs relative to alpine plants by stimulating soil microbial respiration and associated carbon dioxide release. While accelerated microbial respiration can alternatively be a signal of soil carbon accumulation via greater microbial growth (*Liang et al., 2017*), such a mechanism is unlikely to have been responsible here because it would have led to an increase in microbial biomass carbon under lowland plants, which we did not observe.

We speculate that lowland plants accelerated microbial activity in the glasshouse experiment by increasing the quantity of root exudates released into the soil, for four reasons. First, two-pool models of microbial respiration revealed that soils supporting any plant contained a soil carbon pool that was effectively absent from a bare soil control, and which was a factor of 1.5 ± 0.7 (mean ± standard error [SE]) times larger under lowland plants than alpine plants (*Figure 3b*; *Supplementary file 2*; p = 0.046). This carbon pool decayed at a similar rate between treatments (*Figure 3c*; *Supplementary file 2*; p = 0.631), indicating that it differed in size, but likely not composition, between lowland and alpine treatments. Second, the same carbon pool was consumed rapidly by soil microbes (half-life = 5.1 ± 0.4 days), and we also found that lowland plant effects on microbial respiration were lost after 9 days of incubation without plants (*Figure 3a*). Plants thus supplied compounds to microbes that were depleted within this timeframe. While not directly measured here, a 9-day decay period corresponds to the time expected for newly photosynthesized carbon dioxide to be released through

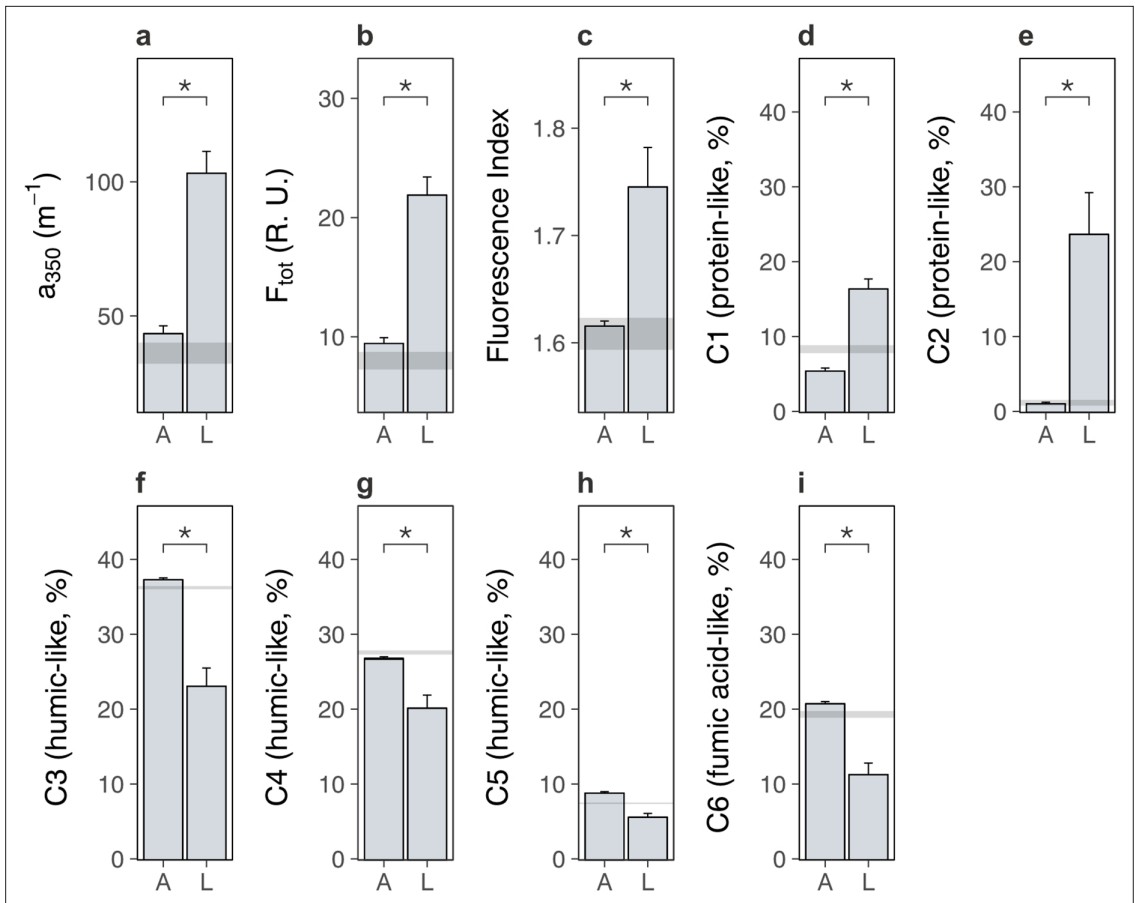

**Figure 4.** Lowland and alpine plants effects on alpine soil dissolved organic matter in the glasshouse experiment. Effects of alpine (A) versus lowland (L) plant treatments on alpine soil dissolved organic matter extracted from soil pore water in a glasshouse experiment (mean ± standard error [SE]; grey ribbons: SEs for bare soil). (**a**) Absorbance ($a_{350}$; m$^{-1}$), (**b**) total fluorescence ($F_{tot}$; Raman Units) and (**c**) fluorescence index (FI) ($N = 20$). (**d–i**) Relative abundance (%) of six dissolved organic matter components (C1–C6), where components and associated compound families were derived from PARAFAC modelling (Materials and methods; *Supplementary file 4*). Treatment effects (p < 0.05) are indicated by asterisks (LMEs; see *Supplementary file 2*).

root exudation and respired by soil microbes (*Hafner et al., 2012*; *Studer et al., 2014*), suggesting that this carbon pool was mostly root exudates. Third, lowland plant effects on the alpine soil system were rapid, occurring within a period of weeks in the glasshouse experiment and months in the field experiments. It is unlikely that lowland plants influenced microbial activity via litter inputs within this timeframe (*De Deyn et al., 2008*; *Cornwell et al., 2008*). Finally, DOM contained a higher relative abundance of protein-like substances than humic-like and fulvic acid-like substances (*Figure 4d–i*; *Supplementary file 2*; p < 0.01 in all cases) under lowland plants than alpine plants, also suggesting that lowland plants caused an enrichment in exudate-derived compounds (*Jones et al., 2009*). While further directed studies are required to resolve whether root exudates are truly involved, our findings collectively suggest that lowland plants have the capacity to increase total root exudation into alpine soil relative to resident alpine plants. In turn, root exudation is a well-established regulator of soil microbial decomposition and the associated release of soil carbon as carbon dioxide (*Allison et al., 2010*; *Melillo et al., 2002*).

One explanation for increased soil carbon outputs under lowland plants is that lowland plants have more biomass than alpine plants and therefore stimulate microbial activity to a greater extent (*Figure 2a–c*). We investigated this possibility by summarizing pot-level traits from the glasshouse experiment using a principal components analysis (PCA) and testing for relationships between resulting axes of trait variation and alpine soil microbial respiration (Materials and methods). We found that descriptors of plant biomass represented the most important axis of trait variation

(*Appendix 1—figure 2*) and were positively related to microbial respiration (*Appendix 1—figure 2a*; PC1: LR = 4.43, *N* = 20, p = 0.035). Thus, greater biomass of lowland plants may have contributed to amplified microbial activity in soils with lowland plants in the glasshouse experiment. However, it is unlikely that increased biomass was solely responsible for lowland plant effects in the field experiments because lowland plant cover was low (mean ± SE relative cover: 4.7% ± 0.7%). Furthermore, in the glasshouse experiment we found that trait variation represented by the second principal axis had opposing effects on microbial respiration between lowland and alpine treatments (*Appendix 1—figure 2b*; treatment × PC2: LR = 5.98, *N* = 20, p = 0.015). Lowland plants thus displayed relationships with the alpine soil system that were qualitatively distinct from resident alpine plants. While focussed work is now needed to pinpoint the physiological mechanism behind these differences, the existence of such a distinction raises the possibility that the arrival of warm-adapted lowland plants in warming alpine ecosystems initiates a change in plant community functioning via changes to traits.

Plants adapted to different temperatures can possess distinct traits that have opposing effects on carbon cycle processes (*Walker et al., 2019*). We therefore tested whether lowland plants in both the glasshouse and field experiments indeed differed from alpine plants in traits that could affect the alpine soil system. Publicly available trait data (*Kattge et al., 2020*) (plant height, leaf area, seed mass, SLA, leaf carbon, leaf nitrogen; Materials and methods) relating to all species in the field experiments revealed that lowland plants on average possessed distinct traits from alpine plants (*Appendix 1—figure 3a*; p = 0.025). This was also the case for measured traits of lowland versus alpine plants under standardized conditions in the glasshouse experiment (multidimensional analyses: *Appendix 1—figure 3b*; p = 0.001). Coupled to this, while lowland plants were present at low abundance in the field experiments (mean ± SE relative cover: 4.7% ± 0.7%), the magnitude of soil carbon loss caused by lowland plant presence was greater in plots where lowland plants were more abundant overall (*Appendix 1—figure 3e*; $F_{1,14}$ = 5.79, p = 0.030). By contrast, we found no evidence that soil carbon loss in the field was alternatively related to individual lowland species identity (*Appendix 1—figure 3c, d*) or the composition of the recipient alpine plant community (*Appendix 1—figure 3f, g*). Based on these observations, we hypothesize that, at least in our experiments, the establishment of warm-adapted lowland plants in warmed alpine ecosystems introduces novel traits into the community that alter plant community functioning and thus carbon cycle processes, with effects that intensify as lowland plants become more abundant in the community (*De Deyn et al., 2008*; *Cornwell et al., 2008*). An alternative explanation is that, trait differences aside, the ability of alpine plants to facilitate soil carbon loss is suppressed when growing in warmer climates to which they are not adapted.

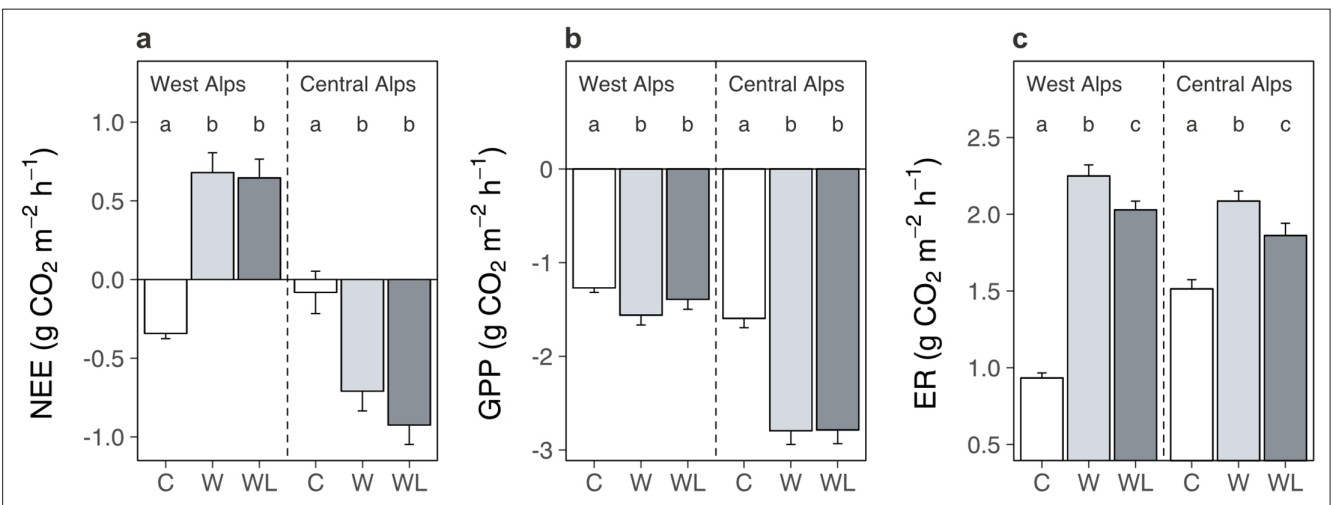

**Figure 5.** Warming and lowland plant effects on alpine ecosystem carbon dioxide fluxes in the field experiment. Effects of transplantation to low elevation (warming, W), transplantation plus lowland plants (warming plus lowlands, WL), or replantation at high elevation (control, C) on alpine ecosystem carbon dioxide fluxes (mean ± standard error [SE]) during one season. (a) Net ecosystem CO₂ exchange (NEE; g CO₂ m⁻² hr⁻¹). (b) Gross primary production (GPP; g CO₂ m⁻² hr⁻¹). (c) Ecosystem respiration (ER; g CO₂ m⁻² hr⁻¹). Measurements were taken in situ at both experiments on multiple dates (western Alps: *N* = 196, central Alps: *N* = 90), with negative values indicating carbon uptake into the ecosystem. Treatment effects (p < 0.05) are indicated by letters (LMEs, Tukey; see *Supplementary file 1*).

While either explanation could ultimately accelerate soil carbon loss from warming alpine ecosystems following the arrival of lowland plants, future studies are needed to fully unravel the environmental dependence of plant trait effects on soil ecosystem processes.

## Lowland plant-induced carbon loss is temporally dynamic

We have shown using two field experiments that lowland plants facilitate a decrease in soil carbon content from warming alpine ecosystems (*Figure 1*) and, with a glasshouse experiment, that they potentially act by accelerating soil carbon outputs via microbial respiration relative to alpine plants (*Figures 3 and 4*), as opposed to decreasing soil carbon inputs. We therefore hypothesized that lowland plant arrival in transplanted alpine plots in the field experiments would drive a net release of carbon dioxide from the ecosystem. As for the glasshouse experiment, we initially examined how lowland plants affected the balance between carbon inputs and outputs of the ecosystem using carbon dioxide flux measurements taken on multiple dates during the growing season (Materials and methods). We then used soil samples collected on one representative date to focus on intrinsic rates (i.e. per unit of biomass) of microbial respiration and growth as processes that promote soil carbon loss and gain, respectively (*Allison et al., 2010*; *Liang et al., 2017*), as well as microbial biomass as the scalar through which these processes affect soil carbon (*Hartley et al., 2008*; *Walker et al., 2018*).

We found that transplantation from high to low elevation (i.e. warming-only) affected alpine ecosystem net carbon dioxide exchange differently between regions (*Figure 5a*; *Supplementary file 1*; p = 0.040). In the western Alps, transplantation changed the ecosystem from a net sink to a net source of carbon dioxide, whereas in the central Alps it made the ecosystem a stronger net carbon dioxide sink. This effect occurred because although transplantation accelerated rates of both gross primary production (*Figure 5b*; *Supplementary file 1*; p < 0.001) and ecosystem respiration (*Figure 5c*; *Supplementary file 1*; p < 0.001), in the western Alps it had stronger effects on ecosystem respiration and in the central Alps stronger effects on gross primary production. Importantly, lowland plants had no significant bearing over net ecosystem exchange (*Figure 5a*), implying that although lowland plants were associated with soil carbon loss from warmed alpine plots (*Figure 1*), this must have occurred prior to carbon dioxide measurements being taken and was no longer actively occurring. Gross primary production in warmed alpine plots was also unaffected by lowland plant presence (*Figure 5b*), which is consistent with the low abundance of lowland plants in alpine communities (mean ± SE relative cover: 4.7% ± 0.7%). As such, lowland plants likely did not facilitate alpine soil carbon loss under warming by decreasing carbon inputs into the ecosystem. By contrast, ecosystem respiration in warmed alpine plots was depressed in the presence versus absence of lowland plants (*Figure 5c*). These findings generally support the hypothesis that lowland plants affect the alpine soil system by changing carbon outputs. However, they contrast with expectations that lowland plants perpetually increase carbon outputs from the ecosystem and thus raise questions about how soil carbon was lost from warmed plots containing lowland plants (*Figure 1*).

Carbon cycle processes are constrained by multiple feedbacks within the soil system, such as substrate availability (*Hartley et al., 2008*; *Walker et al., 2018*) and microbial acclimation (*Crowther and Bradford, 2013*; *Bradford et al., 2019*; *Melillo et al., 2017*), that over time can slow, or even arrest, soil carbon loss (*Walker et al., 2018*; *Bradford et al., 2019*; *Melillo et al., 2017*). We thus interrogated the state of the soil system in the field experiments in the western Alps experiment to explore whether such a feedback may be operating here, in particular to limit ecosystem respiration once soil carbon content had decreased in warmed alpine plots containing lowland plants. We found that biomass-specific rates of microbial respiration in the field were greater following transplantation to the warmer site in the presence, but not absence, of lowland plants (*Figure 6a*; *Supplementary file 1*; p = 0.038). By contrast, biomass-specific rates of microbial growth were slower following transplantation but remained unaffected by lowland plant presence (*Figure 6b*; *Supplementary file 1*; p = 0.016). Thus, despite lower rates of ecosystem respiration overall, alpine soil microbes still respired intrinsically faster in warmed plots containing lowland plants. Moreover, accelerated microbial respiration, but not growth, implies that alpine soils had a higher capacity to lose carbon under warming, but not to gain carbon via accumulation into microbial biomass, when lowland plants were present (*Allison et al., 2010*; *Liang et al., 2017*). These findings align with observations from the glasshouse experiment that lowland plants generally accelerated intrinsic rates of microbial respiration (*Figure 3*), although in field conditions this effect occurred in tandem with warming.

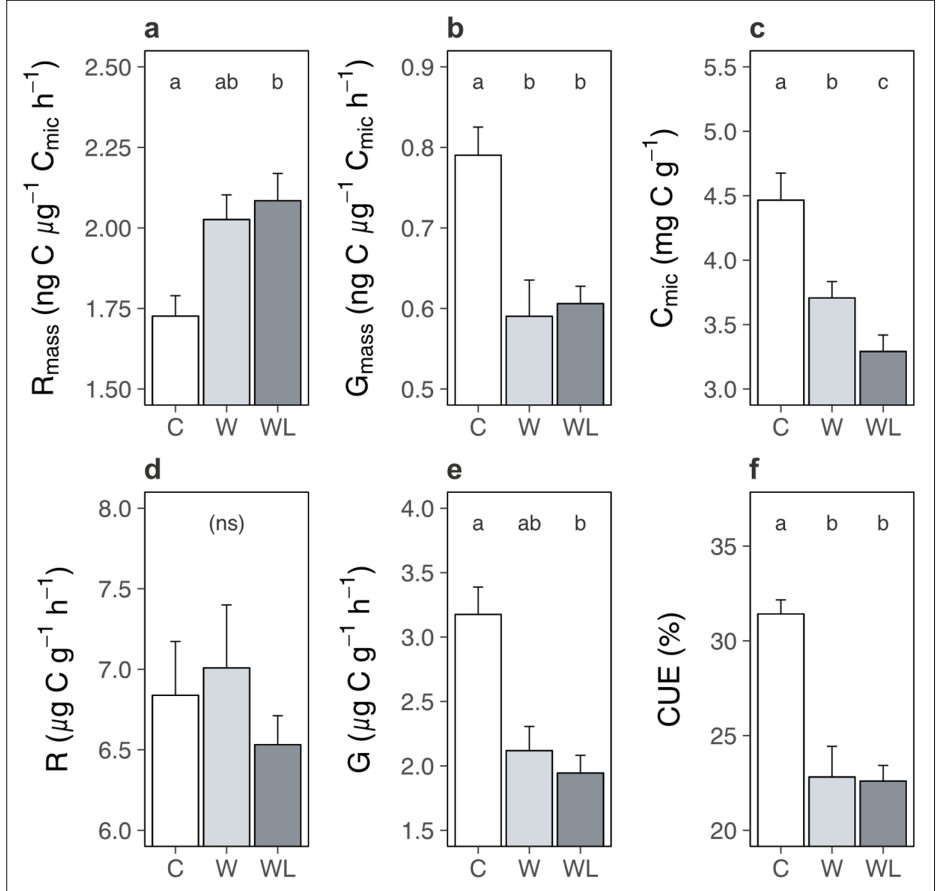

**Figure 6.** Warming and lowland plant effects on the alpine soil system in the field experiment. Effects of transplantation to low elevation (warming, W), transplantation plus lowland plants (warming plus lowlands, WL), or replantation at high elevation (control, C) on alpine soil pools and processes (means ± standard errors [SEs]) after one season. (**a**) Biomass-specific microbial respiration ($R_{mass}$; ng C μg$^{-1}$ C$_{mic}$ hr$^{-1}$). (**b**) Biomass-specific microbial growth ($G_{mass}$; ng C μg$^{-1}$ C$_{mic}$ hr$^{-1}$). (**c**) $C_{mic}$ (μg C g$^{-1}$). (**d**) Total microbial respiration (μg C g$^{-1}$ h$^{-1}$). (**e**) Total microbial growth (μg C g$^{-1}$ h$^{-1}$). (**f**) Microbial carbon use efficiency (CUE; %). Measurements were taken on soils sampled on one date, for (**f**) at both experiments (western Alps: $N = 28$; central Alps: $N = 30$) and for all other variables at the western Alps experiment ($N = 28$). Treatment effects (p < 0.05) are indicated by letters (LMEs, Tukey; see *Supplementary file 1*).

Faster microbial respiration can only yield a loss of soil carbon if it occurs on a per unit of soil basis (*Walker et al., 2018*; *Melillo et al., 2017*; *Prommer et al., 2020*), which was the case in the glasshouse experiment (*Figure 3*). However, in the field we found that lowland plants caused a reduction of microbial biomass carbon in alpine communities transplanted to the warmer site (*Figure 6c*; *Supplementary file 1*; p < 0.001). As such, although microbes generally respired intrinsically faster in the presence of lowland plants, in the field they had become less abundant and so we observed no change to microbial respiration when expressed per unit of soil (*Figure 6d*; *Supplementary file 1*; p = 0.550). The decline in microbial biomass likely took place because soil microbes are chiefly carbon limited (*Hobbie and Hobbie, 2013*), making their abundance tightly coupled to soil carbon content (here: $r_{26} = 0.61$, p = 0.001) (*Walker et al., 2018*). Moreover, since intrinsic rates of microbial growth were unaffected by lowland plant presence, a reduction in microbial biomass translated into a decline in microbial growth per unit of soil (*Figure 6e*; *Supplementary file 1*; *Supplementary file 1*; p = 0.009) – preventing any recovery of microbial biomass via growth (*Liang et al., 2017*). We thus suggest that as soil carbon was lost, microbial biomass also declined, and at the time of measurement this had eliminated the influence of accelerated biomass-specific microbial respiration over the soil system. In short, we propose based on these relationships that the decrease in soil carbon content

observed in warmer transplanted plots containing lowland plants had both started and stopped within one season.

Taken together, one interpretation of our findings is that the establishment of lowland plants in warming alpine ecosystems accelerates intrinsic rates of microbial respiration (*Figures 3 and 6a*), leading to soil carbon release at baseline levels of microbial biomass (*Figures 1 and 3c*), a coupled decline in microbial biomass (*Figure 6c*) and a cessation of further carbon loss from the ecosystem (*Figures 5a and 6d*). Although such a mechanism has been reported in other ecosystems (*Walker et al., 2018*; *Melillo et al., 2017*), applying it here is speculative without additional timepoints because field soil measurements came from a single sampling event after soil carbon had already been lost from the ecosystem. For instance, an alternative mechanism could be that soil microbes acclimate to the presence of lowland plants and this decelerated microbial processes over time (*Crowther and Bradford, 2013*; *Bradford et al., 2019*; *Melillo et al., 2017*). Nevertheless, it remains true that biomass-specific microbial growth and respiration affect the soil system as a function of microbial biomass, and here changes to microbial biomass could explain the behaviour of the soil system in both glasshouse and field experiments. We also found no evidence that soil microbes instead changed their resource use strategy in the presence versus absence of lowland plants (i.e. carbon use efficiency; *Figure 6f*; *Supplementary file 1*; $p < 0.001$; Tukey: $p = 0.996$), pointing away from microbial acclimation as an alternative mechanism. Beyond the mechanism for lowland plant effects on alpine soil carbon loss, it is conceivable that soil carbon loss is not isolated to a single season, but will reoccur in the future even without further warming or lowland plant arrival. This is especially true in the western Alps experiment where warming yielded a net output of carbon dioxide from the ecosystem (*Figure 5a*). Moreover, in our field experiments we simulated a single event of lowland plant establishment and at relatively low abundance in the community (mean ± SE relative cover: 4.7% ± 0.7%), raising the possibility that increases in lowland plant cover or repeated establishment events in the future could facilitate further decreases in alpine soil carbon content under warming.

## General conclusions

In summary, we show that uphill migrations of native lowland plants have the capacity to facilitate warming effects on soil carbon loss from alpine ecosystems. This is important because it demonstrates that plant range expansions between herbaceous plant communities have the potential to contribute to climate feedbacks. These results contrast with expectations from woody species expansions in the Arctic, where shrubs with recalcitrant litter can slow carbon turnover in herbaceous communities and yield a negative climate feedback (*Pearson et al., 2013*; *Cornelissen et al., 2007*). We thus suggest that the influence of range expanding species on ecosystem carbon dynamics depends on the relative trait composition of migrating species versus the recipient community (*Bjorkman et al., 2018*), as well as subsequent effects on environmental processes (*Pearson et al., 2013*). While our findings demonstrate that lowland plants affect the rate of soil carbon release in the short term, short-term experiments, such as ours, cannot resolve whether lowland plants will also affect the total amount of soil carbon lost in the long term. This includes whether processes such as genetic adaptation (in both alpine and lowland plants) (*Walker et al., 2019*) or community change (*Walker et al., 2016*; *Bardgett et al., 2013*) will moderate soil carbon responses to gradual or sustained warming. Nevertheless, here decreases in soil carbon content occurred rapidly despite a low abundance of lowland plants, and increased with total lowland plant cover. Given that both warming and associated plant migrations will intensify over time (*Rumpf et al., 2018*; *Steinbauer et al., 2018*), our estimates of lowland plant impacts may be conservative – particularly if increases in lowland plant cover or repeated establishment events cause recurrent pulses of soil carbon release. Moreover, while alpine ecosystems store only a small fraction of total global soil carbon (*Streit et al., 2014*; *Parker et al., 2015*), redistributions among herbaceous plant communities are occurring globally (*Parmesan, 2006*; *Lenoir and Svenning, 2015*; *Pecl et al., 2017*; *Dolezal et al., 2016*; *Rumpf et al., 2018*; *Steinbauer et al., 2018*), including in similarly cool ecosystems in the Arctic that contain vast soil carbon stocks (*Tarnocai et al., 2009*). This raises the potential for resulting climate feedbacks to also be widespread. Future work should focus on testing the conditions under which this feedback could occur in different mountain regions, as well as other ecosystems experiencing influxes of range expanding plant species, on quantifying how deeply it occurs in shallow alpine soils, and on estimating the magnitude of the climate feedback given both ongoing warming and variation in rates of species range shifts.

## Materials and methods

### Field site description, experimental design

We made use of two independent alpine grassland transplant experiments that were established in Switzerland in the western Alps (46°12′40″N, 7°03′13″E) (*Nomoto and Alexander, 2021*) and central Alps (46°52′20″N, 9°28′58″E) in autumn 2016. The experiments simulate effects of climate warming on alpine ecosystems via transplantation from high to low elevation, including the arrival of lowland plants due to warming-induced uphill plant migrations (*Appendix 1—figure 1*). In the western Alps experiment, the plant community is dominated by *Sesleria caerulea*, *Carex sempervirens*, and *Vaccinium vitis-idea*. In the central Alps experiment the dominant species are *Nardus stricta*, *Alchemilla xanthochlora*, *Trifolium pratense*, and *A. vulneraria*. Broadly, the experiments share the same design. Turfs (area = 1 m², depth = 20 cm) of intact alpine vegetation plus rhizosphere soil were removed from a high-elevation control site (western Alps: 2200 m, MAT = 0.7°C, TAP = 1448 mm; central Alps: 2000 m, MAT = 1.6°C, TAP = 1151 mm; *Karger et al., 2017*) and replanted either at a low-elevation site to simulate climate warming (western Alps: 1400 m, MAT = 6.0°C, TAP = 1236 mm, $N = 10$; central Alps: 1400 m, MAT = 4.4°C, TAP = 1010 mm, $N = 20$; *Karger et al., 2017*) or back into a high-elevation site as a transplantation control (see below). At the low-elevation site, we created an additional treatment to simulate the arrival of lowland plants under warming by planting a low abundance of lowland plant species in a grid design (13 cm between individuals), creating a lowland plant density of approximately 56 individuals per m² (4.7% ± 0.7% total cover) in a background alpine plant community. Lowland plants were positioned randomly in the grid, but with the same number of individuals per species per plot, and we controlled for planting disturbance in all remaining plots using alpine plants in place of lowland plants (see below). In the western Alps region, lowland plant species were *Achillea millefolium*, *Bellis perennis*, *B. erectus*, *Dactylis glomerata*, *M. lupulina*, *Plantago media*, and *S. pratensis*, and individuals were planted into one side of each plot in a split-plot design ($n = 10$). In the central Alps experiment, lowland plant species were *Brachypodium pinnatum*, *Carex flacca*, *Carum carvi*, *D. glomerata*, *Hypericum perforatum*, *Plantago lanceolata*, *Primula veris*, *Ranunculus bulbosus*, *S. pratensis*, *Silene vulgaris*, *Trifolium montanum*, and *Viola hirta*, and individuals were planted into half of the transplanted turfs ($n = 10$). While exact dispersal distances for selected lowland species are unknown, all species are widespread and are expected to migrate uphill under warming (*Rumpf et al., 2018*) and the horizontal distance between high and low sites in the field experiments was always less than 2 km. In the western Alps experiment, lowland individuals were pre-grown in a greenhouse in steam-sterilized field soil and planted into the plots in May 2017. In the central Alps experiment, lowland individuals were taken from the surrounding vegetation and planted into the plots in autumn 2016. In both regions, soil was partially removed from lowland individuals prior to planting, and we performed an additional experiment to eliminate the possibility that small contaminations of lowland soil biota during planting could have modified alpine soil microbial community composition (see Field experiment soil biota validation). Overall, the experimental design in both regions yielded three treatments ($n = 10$ for each) arranged in a block design: transplantation from high to low elevation (i.e. warming-only, W); transplantation plus low-elevation plants (i.e. warming plus lowland plant establishment, WL); and replantation at the high-elevation origin site (i.e. control, C). One turf at the low-elevation site in the western Alps experiment failed, leaving a total sample size of $N = 28$ and $N = 30$ in the western Alps and central Alps, respectively.

### Field experiment controls and validation

We controlled for transplantation disturbance in both experiments by moving alpine control turfs back into a high-elevation control site. In the western Alps experiment, alpine turfs were planted into different positions at their origin high-elevation site. In the central Alps experiment, alpine turfs were planted into a new high-elevation site at the same elevation. As such, we eliminated the possibility that alpine turfs changed simply because they were disturbed or were moved to a different location (i.e. irrespective of elevation). The transplantation process had no effect on alpine soil carbon content, in that we found no difference in soil carbon content between replanted alpine control turfs at high-elevation sites and undisturbed alpine soil in the surrounding area (LR = 2.27, $N = 16$, p = 0.132). We also controlled for potential effects of planting disturbance in all turfs not planted with lowland plants (i.e. remaining turfs at low elevation and all control turfs at high elevation). In the western Alps experiment, we planted alpine species from the high-elevation control site into the other side of

alpine turfs at the low-elevation site, as well as into all control turfs at the high-elevation site. This was done in autumn 2016 using the same approach and randomized hexagonal grid design. In the central Alps experiment, we mimicked planting disturbance by digging up and replanting existing individuals in alpine turfs at the low-elevation site not containing lowland plants, as well as all control turfs at the high-elevation site. This was done in spring 2017 using the same grid design. It is important to note that we did not perform a reverse transplantation (i.e. from low to high elevation), so we cannot entirely rule out the possibility that transplantation of any community to any new environment could yield a loss of soil carbon. Nevertheless, it is well established that climate warming promotes soil carbon loss (*Melillo et al., 2002*; *Crowther et al., 2016*; *Davidson and Janssens, 2006*), and we found that soil from low-elevation sites (i.e. the potential end-member for warming alpine ecosystems) contained less carbon than soil from the alpine sites (70.1 ± 3.05 mg C g$^{-1}$ soil dry mass compared to 145.0 ± 5.29 mg C g$^{-1}$ soil dry mass; LR = 37.2, $N$ = 27, p < 0.001), validating our expectation that alpine soil would lose carbon under warming. Finally, while small variations in experimental design between regions, such in the order of planting in W and WL treatments, likely introduced some error when comparing experiments, we used both regions together so that we could identify treatment effects that were consistent among them and were thus not artefacts of experimental design.

## Field experiment soil biota validation

We tested for the possibility that soil carbon loss was alternatively caused by low-elevation soil microbes hitchhiking on lowland plants during planting. Specifically, we performed an experiment at the central Alps site to determine whether inoculating alpine plots with low-elevation soil biota affected soil microbial community composition. We first generated a low-elevation soil biota inoculum from rhizosphere soil from the same area where lowland plants were sourced and gently mixing and sieving it with tap water (following refs *van de Voorde et al., 2012*; *De Vries et al., 2015*). We then added 3 L of inoculum to one half of each of 10 × 1 m$^2$ alpine turfs equivalent to those in the main experiment, followed by 3 L of water to rinse the inoculum from the vegetation and facilitate its infiltration into the soil. The other half of the plot (separated by an impermeable rubber barrier buried 15 cm deep into the soil around the perimeter of each subplot) received 6 L of water as a control. Soil inoculation was performed in mid-May and repeated in mid-June 2017. In October 2017, we sampled soils by pooling three soil cores (ø = 1 cm, $d$ = 5 cm) from each subplot, following which we determined soil microbial community composition by sequencing the 16S rRNA gene for bacteria/archaea and the ITS1 region for fungi (*Herbold et al., 2015*). DNA was extracted (NucleoSpin Soil kit, Macherey-Nagel, Düren, Germany) and amplified using quadruplicate PCRs, alongside extraction and PCR blank controls. PCR products were purified (MinEluteTM PCR purification kit, Qiagen, Hilden, Germany) prior to sequencing on a MiSeq sequencing platform (Illumina, San Diego, USA). Sequence data were processed using the OBITools software package (*Boyer et al., 2016*) and following *Zinger et al., 2017*. Paired-end reads were merged, assigned to their respective samples and dereplicated, and sequences with too short read lengths and with only one read were excluded. Remaining sequences were clustered using the SUMACLUST (*Mercer et al., 2013*) algorithm and summarized into operational taxonomic units (OTUs) using the Infomap Community Detection Algorithm (*Rosvall and Bergstrom, 2008*), following which we calculated per-sample relative OTU abundances for bacteria/archaea and fungi separately. We found that the alpine soil microbial community was resistant to the introduction of low-elevation soil biota, in that inoculation had no effect on soil bacterial (*Appendix 1—figure 4a*; PERMANOVA; $F_{1,9}$ = 0.75, p = 0.573) or fungal (*Appendix 1—figure 4b*; $F_{1,9}$ = 0.75, p = 0.774) community composition. The inoculation represented a strong soil biota treatment relative to contamination by hitchhiking microbes. As such, even if some contamination by low-elevation soil biota did occur, it was unlikely to have resulted in a change to the extant soil community, ruling out the possibility that contamination was alternatively responsible for the soil carbon loss observed when lowland plants were present in transplanted alpine plots.

## Field plant community composition

We surveyed the composition of alpine communities in summer 2017. This was done by dividing each plot into 5 × 5 cm cells and visually estimating the canopy cover of all species rooted in every grid cell. Species cover was approximated with an ordinal scale in which the first three cover categories correspond to 1/16, 1/8, and 1/4 of the grid cell area (25 cm$^2$), and subsequent cover categories

were constant increases of 1/4 of the grid cell area. Plants were identified to species level whenever possible, although some taxa were identified only to genus or subfamily levels when species identity was unclear based on vegetative traits. Species cover ($cm^2$) was then calculated as the species-wise sum of cover estimates across all grid cells in the quarter of the plot used for soil and ecosystem respiration measurements (see below).

## Field soil carbon content, microbial physiology, microbial biomass

We collected soil samples from all treatments in both field experiments ($N = 58$) in August 2017. In each plot, we created a composite sample from three cores (ø = 1 cm, approx. $d$ = 7 cm) no closer than 7 cm from a planted individual and from the same quarter of the plot used for ecosystem respiration measurements (see below; *Appendix 1—figure 1*). We focused on rhizosphere soil for two reasons. First, alpine soils are shallow and alpine plants are shallow-rooting, making the rhizosphere the most logical target for a first assessment of warming and lowland plant effects on alpine soil carbon cycling. Second, we wanted to avoid any potential contamination from the base of the turf (depth = 20 cm). Soils were kept cool, transported immediately to the laboratory, sieved (2 mm mesh size), adjusted to 60% of water holding capacity (WHC) and pre-incubated for 3 days at field temperature prior to measurements. We measured microbial biomass carbon (mg per g dry mass) via chloroform fumigation extraction followed by analysis on a TOC-VCPH/CPNTNM-1 analyzer (Shimadzu; 48 hr incubation period; 1 M KCl extraction for fumigated and non-fumigated samples; conversion factor 0.45) (*Vance et al., 1987*). While this approach also yielded measurements of dissolved organic carbon (DOC; mg C per g dry mass), treatment effects on this variable echoed those of microbial biomass carbon (*Supplementary file 1*) so we did not interpret them separately. Soil organic carbon content (%) was measured on dry soil (60°C for 48 hr) with a Carlo Erba 1,110 elemental analyzer (EA, CE Instruments) coupled to a Delta Plus IRMS via a Conflo III (Thermo Fisher). It was not possible to take widespread measurements of soil bulk density due to the destructive sampling required while other studies were underway (e.g. *Nomoto and Alexander, 2021*). Instead, we took additional soil cores (ø = 5 cm, $d$ = 5 cm) from the central Alps experiment in 2021 once other studies were complete to indirectly explore whether lowland plant effects on soil carbon content in warmed alpine plots could have occurred due to changes in soil bulk density. We found that although transplantation to the warmer site increased alpine soil bulk density (LR = 7.18, p = 0.028, Tukey: p < 0.05), lowland plants had no effect (Tukey: p = 0.999). It is not possible to make direct inferences about the soil carbon stock using measurements made on different soil cores 4 years apart. Nevertheless, these results make it unlikely that lowland plant effects on soil carbon content in warmed alpine plots occurred simply due to a change in soil bulk density. We additionally quantified microbial growth, respiration and carbon use efficiency in soils from the western Alps experiment ($N = 28$) using the incorporation of $^{18}O$ into DNA (*Walker et al., 2018*). Briefly, we incubated 500 mg of soil for 24 hr at field temperature following addition of either $^{18}O$-$H_2O$ to 20 atom% enrichment and 80% of WHC or the same volume of molecular grade non-labelled $H_2O$ as a natural abundance control. Gas samples were taken at the start and end of the 24 hr incubation and analyzed for $CO_2$ concentration with a Trace GC Ultra (Thermo Fisher) to determine microbial respiration (μg C per g dry mass per hr). At the end of the incubation, soil was snap frozen in liquid nitrogen, and DNA was extracted (FastDNA SPIN Kit for Soil, MP Biomedical), quantified (Quant-iT PicoGreen dsDNA Assay Kit, Thermo Fisher) and analyzed for $^{18}O$ abundance and total oxygen content using a thermochemical EA coupled to a Delta V Advantage IRMS via a Conflo III (Thermo Fisher). Following *Walker et al., 2018*, we calculated total DNA production during the 24-hr incubation (μg DNA per g dry mass) and coupled it with measures of microbial biomass carbon content to derive microbial growth (μg C per g dry mass per hr) and carbon use efficiency (%).

## Field ecosystem respiration

We measured rates of ecosystem respiration (g $CO_2$ per $m^2$ per h) on multiple dates (western Alps: seven dates; central Alps: three dates) between July and August 2017. One month prior to the first measurement, we installed permanent gas sampling collars (h = 10 cm, ø = 30 cm) in each plot to a depth of 5 cm to create an airtight seal at the soil surface (*Walker et al., 2019*). Measurements were made by enclosing plants and soil within the collars with an opaque chamber (h = 50 cm, ø = 30 cm) and, following a 30-s equilibration period, monitoring the increase in $CO_2$ concentration at 5-s intervals over a 60-s period (Vaisala GMP343 $CO_2$ probe attached to an HM70 meter, Vaisala). Chambers

contained a fan to mix the headspace during measurements and a small sealable hole on the top that was opened when placing the chamber on sampling collars to avoid pressurization inside the chamber. Soil temperature and moisture were recorded for each measurement and used alongside measurements of air temperature, air pressure, enclosure volume, and soil surface area to calculate the $CO_2$ flux (*Walker et al., 2019*). Where necessary, we repeated measurements in the field to attain a visually consistent $CO_2$ change over time, and during data processing we discarded fluxes with an *r*-squared of less than 0.9 for this relationship.

## Glasshouse experiment design

We established a 6-week pot experiment to test for effects of lowland versus alpine plants on the alpine soil system. Specifically, we filled pots (ø = 10 cm, depth = 9 cm) with sieved soil (mesh size 4 mm) from the high-elevation site of the western Alps field experiment, and planted them with six lowland species (*B. erectus*, *S. pratensis*, *M. lupulina*) or six alpine species (*P. alpinum*, *A. vulneraria* ssp. *alpestris*, *P. alpina*) to create a lowland only treatment, an alpine only treatment and a bare soil control (*N* = 30). We selected a grass, herb, and legume species for each treatment, all of which were found naturally at the low (lowland treatment) or high (alpine treatment) elevation sites in the western Alps field experiment. Vegetated treatments contained two individuals of all lowland or alpine species planted in a circular pattern such they were never adjacent to conspecifics or plants of the same functional type (i.e. grass, herb, legume), making replicates within treatments identical in species composition and neighbourhoods. The original design also included a mixed community treatment containing one individual of each lowland and alpine species. However, the mixed treatment had double the species richness of other vegetated treatments and contained an equal cover of lowland and alpine plants (compared to <5% lowland plant cover in field plots), so we focussed our analyses on the comparison between alpine and lowland plant treatments to understand how lowland plants affect the soil differently from alpine plants. All plants were pre-grown from seed provided by commercial suppliers (https://www.hauenstein.ch, https://www.schutzfilisur.ch, https://www.saatbau.at) in steam-sterilized potting soil for 2–3 weeks to approximately the same biomass and root-washed immediately prior to planting. Pots were arranged in a randomized block design (*n* = 10) in a greenhouse under standardized growing conditions (22°C; 50% humidity; no artificial light), and were watered every 2–3 days as needed. After 6 weeks, we destructively harvested the plants and soil to characterize treatment effects on plant physiology and the soil system.

## Glasshouse plant physiology

We measured maximum leaf-level photosynthesis (photosynthetic capacity; $A_{max}$; µmol $CO_2$ per $m^2$ per s) and stomatal conductance ($g_s$; µmol $H_2O$ per $m^2$ per s) under optimum conditions on one individual per species per pot using an open path infrared gas analyzer connected to a 2.5-cm chamber (CIRAS-2 and PLC-6, PP Systems) (*Walker et al., 2019*). Readings were taken once for each individual and corrected using measurements of leaf area ($cm^2$) on the same leaf, which we also dried and weighed to determine SLA ($cm^2$ per g). Following soil pore water sampling (below), we quantified total, aboveground and belowground biomass of all individuals in the experiment by destructively sampling, washing, drying (60°C for 48 hr), and weighing all plant tissue.

## Glasshouse soil pore water analysis

We collected soil pore water from all pots prior to soil sampling (see below) to determine treatment effects on DOM quality. Soil water was sampled by leaching pots twice with 50 mL sterile tap water, combining and filtering leached water (filter pore size 0.2 µm) and measuring absorbance (200–900 nm with a 1-nm increment; 1-cm quartz cuvette; UV–vis spectrophotometer, Shimadzu) and fluorescence intensity (excitation: 240–450 nm with a 5-nm increment; emission: 290–550 nm with a 2.5-nm increment; Fluorolog-3 spectrometer, Horiba). We corrected absorbance spectra against a milli-Q water blank, calculated Napierian absorption coefficients and used these data to derive absorbance at 350 nm ($a_{350}$; a proxy for DOM content). We expressed fluorescence data as total fluorescence ($F_{tot}$; Raman Units) and the FI, that is the ratio of emission intensity at 470 versus 520 nm at an excitation wavelength of 370 nm (a proxy for microbially derived DOM) (*McKnight et al., 2001*). We also derived excitation–emission matrices (EEMs) to quantify the relative abundance of different components of DOM through PARAFAC algorithms (*Stedmon et al., 2003*). Following preprocessing (i.e. Raman

scattering removal and standardization to Raman Units) and normalization (*Zepp et al., 2004*), we used the drEEM toolbox (*Murphy et al., 2013*) in MATLAB (Mathworks) to obtain a six-component PARAFAC model. The model was validated with split-half analysis, random initialization and visualization of EEM residuals, after which we calculated the relative abundance of each component as the maximum fluorescence signal of the component against the total maximum fluorescence signal of all components. Finally, we classified components against classical EEM nomenclature (*Supplementary file 4*; *Fellman et al., 2010*; *Coble et al., 2014*). While PARAFAC modelling provides a means to discriminate between DOM sources (e.g. components C3 and C4; *Supplementary file 4*), we did not physically separate plant- and microbe-derived DOM when sampling soil pore water. As such, observed changes to DOM composition were likely a result of changes to both plant and microbial processes (see Main text).

## Glasshouse soil carbon pools, microbial respiration

We took soil cores (ø = 1 cm, *d* = 7 cm) from the centre of each pot (i.e. equidistant from all individuals) to quantify soil carbon pools and microbial respiration. Soils were sieved (2-mm mesh size), adjusted to 60% of their WHC and pre-incubated at 22°C for 3 days prior to measurements. We quantified concentrations of microbial biomass carbon in all soils as for field soil (see above; mg C per g dry mass). We determined microbial respiration by incubating 8 g sieved soil in the dark at 22°C for a further 6-week period without plants. We measured microbial respiration daily for the first 4 days, then approximately biweekly until the end of the 6-week period. We measured $CO_2$ concentrations (EGM-4, PP Systems) of gas samples taken approximately 24 hr after flushing enclosed vials (*v* = 126 ml) with $CO_2$-free air, and calculated microbial respiration (µg C per g dry mass per hr) using accompanying measurements of enclosure volume, air pressure, and soil mass (*Walker et al., 2019*). After each measurement, soils were weighed and, when necessary, re-adjusted to 60% of their WHC.

## Two-pool mixed models of soil carbon mineralization

We described the kinetics of soil carbon mineralization during the follow-up incubation of glasshouse soil by employing a two-pool mixed model to microbial respiration data (*Bonde and Rosswall, 1987*). Specifically, we constructed a model that considered a first pool that followed first-order decomposition kinetics (i.e. a small soil carbon pool with a fast turnover) and a second pool that followed zero-order kinetics (i.e. decay rate was constant over the incubation period, representing a large soil carbon pool with a slow turnover):

$$C_{min} = C_l \left( 1 - e^{-tk_l} \right) + B \cdot t$$

where $C_{min}$ is the cumulative $CO_2$ respired (µg C $g^{-1}$) over the incubation time *t* (d), $C_l$ and $k_l$ are the size (µg C $g^{-1}$), and decomposition constant ($d^{-1}$) of the first-order kinetics carbon pool, respectively, and *B* is the basal respiration rate (µg C $g^{-1}$ $d^{-1}$) of the zero-order kinetics carbon pool. Given that $C_{min}$ and *t* were known, we ran the two-pool mixed model with the nlsList function (maxiter = 100, tol = 1 × $10^{-5}$, minFactor = 1/1024) in *R Development Core Team, 2014* using the package 'nlme' (*Pinheiro et al., 2019*) for each pot separately to estimate values for $C_l$, $k_l$, and *B*. Models failed to converge for one sample, and we excluded four additional samples for which models made negative pool estimates (*N* = 25).

## Statistical analysis

The majority of statistical test outputs appear as p values in the Main text, with full outputs being provided in *Supplementary files 1–3*. See text below for exceptions. Data analyses and visualization were undertaken in *R Development Core Team, 2014* using the packages 'cowplot' (*Wilke, 2019*), 'drake' (*Michael Landau, 2018*), 'emmeans' (*Lenth, 2020*), 'nlme' (*Pinheiro et al., 2019*), 'tidyverse' (*Wickham et al., 2019*), and 'vegan' (*Oksanen, 2019*). Where necessary, region was accounted for in statistical models as a fixed effect, since two levels was too few to be included as a random intercept term. As such, region effects are presented (*Supplementary files 1 and 3*), but not interpreted. Statistical methods are described as they appear in the Main text.

## Warming and warming plus lowland plant effects on soil carbon content

We used field data to test for treatment (C, W, WL) effects on soil carbon content using an LME model including region as a fixed effect and block as a random intercept term (see *Supplementary file 1* for test output). We also calculated how much more soil carbon loss occurred in the presence versus absence of lowland plants by fitting a single LME testing for effects of soil temperature (plot means of all sampling dates), the presence or absence of lowland plants and their interaction on soil carbon content (see *Appendix 1—figure 3* for test outputs). In order to compare slopes between C–W and C–WL treatment combinations, we duplicated C treatment values, assigned one set to each C–W and C–WL pair and accounted for this using a random intercept term. We also included region as a random slope term. We then tested how much more soil carbon loss would occur per °C of warming when lowland plants are present by calculating the relative difference (% ± 95% CIs) between the C–WL and C–W slopes.

## Plant mechanisms for soil carbon loss

We expressed soil carbon content as the block-wise percent changes between WL and W treatments to derive a metric for lowland plant-induced soil carbon loss (i.e. % soil carbon remaining). We tested for lowland species identity effects on soil carbon remaining using separate LMEs for each region (due to different lowland species pools) that tested for individual species cover ($cm^2$) effects on soil carbon remaining in WL plots, including species as a random intercept term (see *Appendix 1—figure 3* for test outputs). We also performed separate linear models including species as fixed effects, none of which were statistically significant (*Supplementary file 3*). We tested for effects of total lowland species cover ($cm^2$) on soil carbon remaining in WL plots using a linear model also including region as a fixed effect. We summarized alpine community composition in W and WL treatments using an NMDS (Hellinger distance matrix, two axes), following which we expressed NMDS scores as the block-wise percent changes between W and WL treatments to derive metrics for lowland plant-induced alpine plant community change, and used a linear model that tested for effects of changes to NMDS scores on soil carbon remaining in WL plots including region as a fixed effect (see *Appendix 1—figure 3* for test outputs). For reference, raw NMDS scores did not significantly differ between W and WL treatments (LME: NMDS #1: LR = 1.57, p = 0.210; NMDS #2: LR = 2.67, p = 0.102). Finally, we tested for differences between alpine and lowland species traits in the field experiment using a PERMANOVA on a PCA of species mean trait values ($log_{10}$ plant height, $log_{10}$ leaf area, $log_{10}$ seed mass, SLA, leaf carbon, leaf nitrogen) from the TRY database (*Kattge et al., 2020*) for all species present (where necessary, missing values were gap-filled following *Schrodt et al., 2015*), and in the glasshouse experiment using a PERMANOVA on a PCA of physiological measurements (total biomass, leaf biomass, root biomass, SLA, $A_{max}$, $g_s$) for all plants (see *Appendix 1—figure 3* for test outputs). Traits were selected to explore broad life history differences between alpine and lowland plant species (*Diaz et al., 2016*), but were not used to identify the specific trait(s) responsible for lowland plant effects on alpine soil carbon loss. We used TRY traits for field species due to the large number of species present in the communities (*N* = 118), but used measured traits in the glasshouse experiment to provide direct evidence of physiological differences between alpine and lowland species.

## Lowland versus alpine plant effects on the soil system

We used the glasshouse experiment to test for lowland versus alpine plant treatment effects on soil microbial biomass carbon, the size ($C_l$) and decay rate ($k_l$) of the first modelled soil carbon pool and all variables relating to soil DOM quality using LMEs including block as a random intercept term. We tested for effects of treatment, time, and their interaction on soil microbial respiration using the same LME structure but including the bare soil treatment. In all cases, including/excluding the bare soil treatment had no effect over model outcomes, but mean ± SE values for this treatment are included in figures for reference. Finally, we explored how plant physiology in lowland and alpine treatments at the pot level affected initial rates of soil microbial respiration. We defined initial microbial respiration as the pot-wise mean respiration rates of the first 9 days of soil incubation (i.e. while treatment effects

persisted; see *Figure 3a*), and expressed pot-level plant physiology using scores from the first two axes of a PCA performed on sums of total, root and shoot biomass and biomass-weighted means of SLA, $A_{max}$, and $g_s$. We then tested for effects of treatment, PC1 and PC2 scores and their interaction on initial microbial respiration using an LME including block as a random intercept term. Statistical test outputs for LMEs are provided in *Supplementary file 2*, except for treatment and plant physiology effects on microbial respiration, which appear in the Main text.

## Sequence of warming and lowland plant effects on alpine soil carbon loss

We used field data to test for treatment (C, W, WL) effects on soil microbial biomass carbon and soil DOC using LME models including region as a fixed effect and block as a random intercept term. We tested for treatment effects on ecosystem respiration using an LME that included region as a fixed effect and sampling date nested in block as a random intercept term. We tested for treatment effects on soil microbial growth and respiration (per unit of soil and per unit of microbial biomass) and carbon use efficiency in the western Alps experiment using LMEs including block as a random intercept term. Statistical test outputs for these variables are provided in *Supplementary file 1*. We also used Pearson Product Moment correlations to test for associations between soil microbial biomass and soil carbon content, microbial respiration and growth in the western Alps experiment, irrespective of experimental treatment.

### Data availability

Data supporting the findings of this study are stored on the Open Science Framework website (https://www.osf.io; DOI: https://doi.org/10.17605/OSF.IO/S54CH). The R code used to process and analyse the data are available in a Github repository with the URL: https://github.com/tom-n-walker/uphill-plants-soil-carbon (copy archived at swh:1:rev:a9e7a872d22a45dbd91bb00751d522812138eefd; *Walker, 2022*).

### Acknowledgements

JMA received funding from the European Union's Horizon 2020 research and innovation program (grant no. 678841). JMA and TWNW received funding from the Swiss National Science Foundation (grant no. 31,003 A-176044). KG is funded by the Swiss National Science Foundation (grant no. PZ00P2-174047). SB and the central Alps experiment were funded by the Swiss National Science Foundation (grant no. 31,003 A-173210). TM received funding from the French National Research Agency (ANR-20-CE02-0021).

## Additional information

### Funding

| Funder | Grant reference number | Author |
| --- | --- | --- |
| European Union Horizon 2020 | 678841 | Jake M Alexander |
| Swiss National Science Foundation | 31003A-176044 | Jake M Alexander<br>Tom WN Walker |
| Swiss National Science Foundation | PZ00P2-174047 | Konstantin Gavazov |
| Swiss National Science Foundation | 31003A-173210 | Sebastián Block |
| French National Research Agency | ANR-20-CE02-0021 | Tamara Münkemüller |

The funders had no role in study design, data collection, and interpretation, or the decision to submit the work for publication.

## Author contributions
Tom WN Walker, Conceptualization, Data curation, Formal analysis, Funding acquisition, Investigation, Methodology, Project administration, Validation, Visualization, Writing – original draft, Writing – review and editing; Konstantin Gavazov, Pierre Mariotte, Constant Signarbieux, Hanna Nomoto, Data curation, Investigation, Writing – review and editing; Thomas Guillaume, Thibault Lambert, Data curation, Formal analysis, Investigation, Writing – review and editing; Devin Routh, Data curation, Formal analysis; Sebastián Block, Tamara Münkemüller, Conceptualization, Data curation, Investigation, Writing – review and editing; Thomas W Crowther, Data curation, Formal analysis, Writing – review and editing; Andreas Richter, Methodology, Project administration, Resources, Writing – review and editing; Alexandre Buttler, Conceptualization, Project administration, Resources, Writing – review and editing; Jake M Alexander, Conceptualization, Funding acquisition, Investigation, Project administration, Resources, Writing – original draft, Writing – review and editing

## Author ORCIDs
Tom WN Walker http://orcid.org/0000-0001-8095-6026
Konstantin Gavazov http://orcid.org/0000-0003-4479-7202
Pierre Mariotte http://orcid.org/0000-0001-8570-8742

## Decision letter and Author response
Decision letter https://doi.org/10.7554/eLife.78555.sa1
Author response https://doi.org/10.7554/eLife.78555.sa2

# Additional files

## Supplementary files
• Supplementary file 1. Table showing linear mixed-effects model outputs for effects of field treatment (control, warming, warming plus lowland plants), region and their interaction on soil variables. LR: likelihood ratio.

• Supplementary file 2. Table showing linear mixed-effects model outputs for treatment (alpine plants, lowland plants) effects on plant traits and soil variables in the glasshouse experiment. LR: likelihood ratio.

• Supplementary file 3. Table showing statistical test outputs for lowland species identity effects on soil carbon loss.

• Supplementary file 4. Table showing DOM components identified through PARAFAC modelling.

• MDAR checklist

## Data availability
All data contributing to the findings of this study have been deposited in the OSF under the DOI: https://doi.org/10.17605/OSF.IO/S54CH. The R code used to process and analyse the data are available in a Github repository with the URL: https://github.com/tom-n-walker/uphill-plants-soil-carbon (copy archived at swh:1:rev:a9e7a872d22a45dbd91bb00751d522812138eefd).

The following dataset was generated:

| Author(s) | Year | Dataset title | Dataset URL | Database and Identifier |
|-----------|------|---------------|-------------|-------------------------|
| Walker TWN | 2021 | Data: uphill plant migrations soil carbon loss | https://doi.org/10.17605/OSF.IO/S54CH | Open Science Framework, 10.17605/OSF.IO/S54CH |

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

## Appendix 1

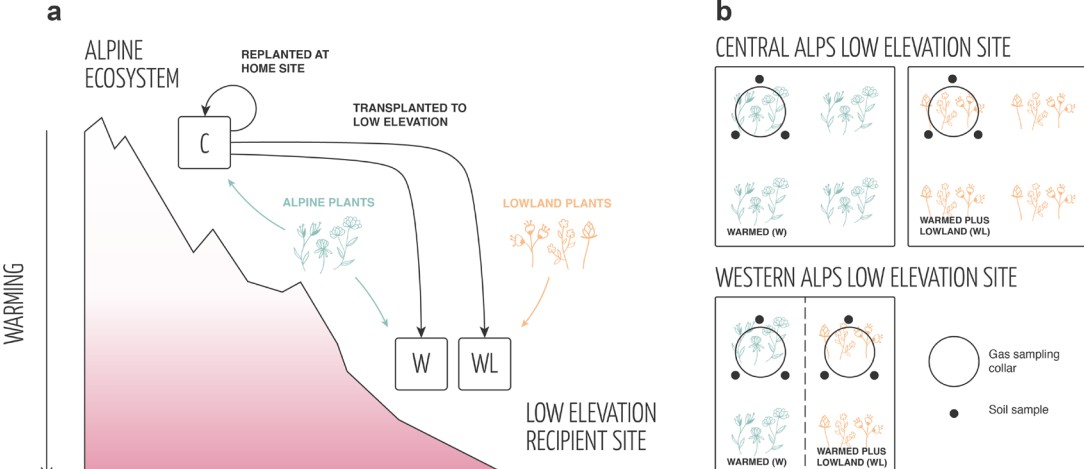

**Appendix 1—figure 1.** Experimental design. (**a**) In both field experiments, intact alpine turfs containing alpine plant communities plus rhizosphere soil were removed from their high-elevation home sites and either replanted at the home site (negative control; C) or transplanted to a low-elevation recipient site to simulate climate warming (W, WL). At low elevation, half of the turfs were planted with a low abundance of lowland plant species to additionally simulate the arrival of migrating plant species in the ecosystem (WL). Both other treatments (i.e. C, W) were planted with the same abundance of alpine plant species as a disturbance control. (**b**) The layout of W and WL treatments at the low-elevation site differed between western and central alps experiments. In the central alps, W and WL treatments were established in distinct turfs planted adjacently in a block design (*N* = 10). In the western alps, W and WL treatments were established in different halves of the same turf in a split-plot design (*N* = 10).

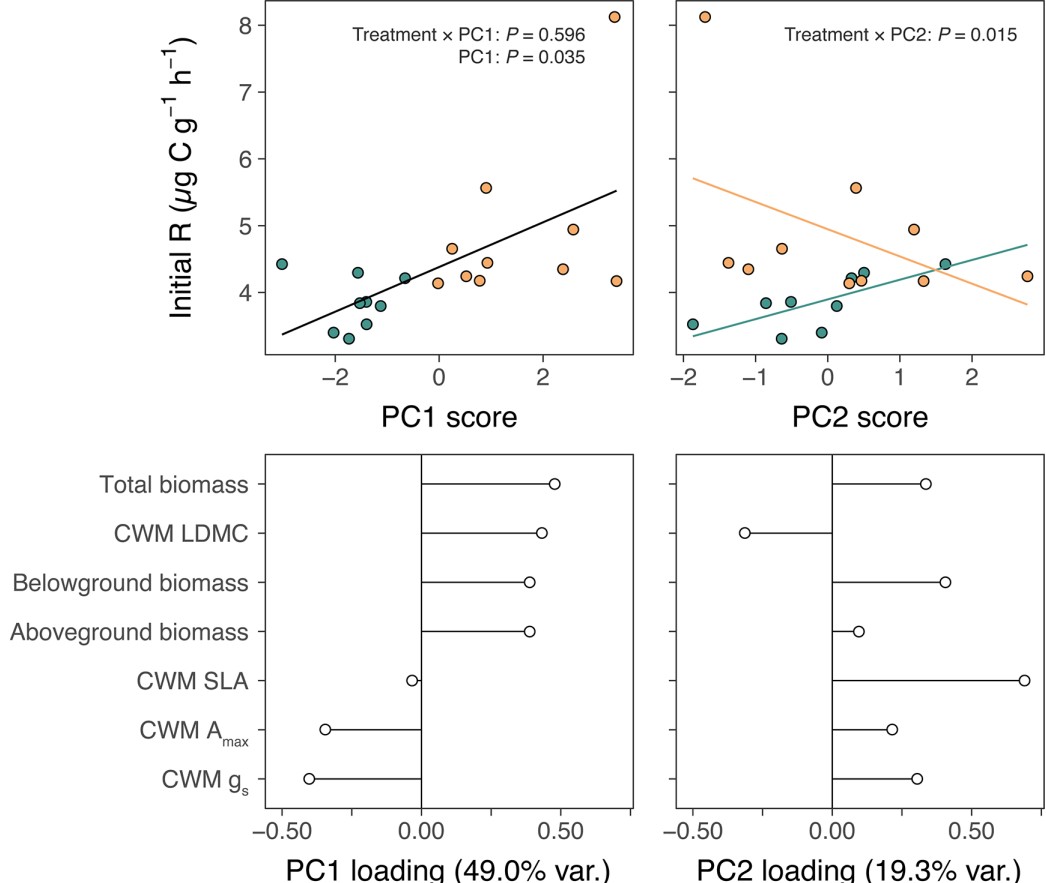

**Appendix 1—figure 2.** Alpine and lowland plant effects on alpine soil microbial respiration. Relationships between plant physiology, expressed as (**a**) PC1 and (**b**) PC2 scores from PCAs of pot-level measurements, and rates of soil microbial respiration during the first nine days of incubation (initial R; μg C g$^{-1}$ h$^{-1}$) in alpine (green) versus lowland (yellow) treatments of the glasshouse experiment. Fit lines describe relationships as determined by a single linear mixed-effects model including treatment, PC1, PC2, and treatment × PC# interactions (N = 20; Main Text), with different colours in (**b**) indicating a significant treatment × PC2 interaction. Loadings of plant physiological measurements on (**c**) PC1 and (**d**) PC2 axes. Total, belowground, and aboveground biomass are sums of all plants per pot and specific leaf area (SLA), maximum photosynthetic capacity ($A_{max}$), and stomatal conductance ($g_s$) are pot-wise community-weighted mean (CWM) values.

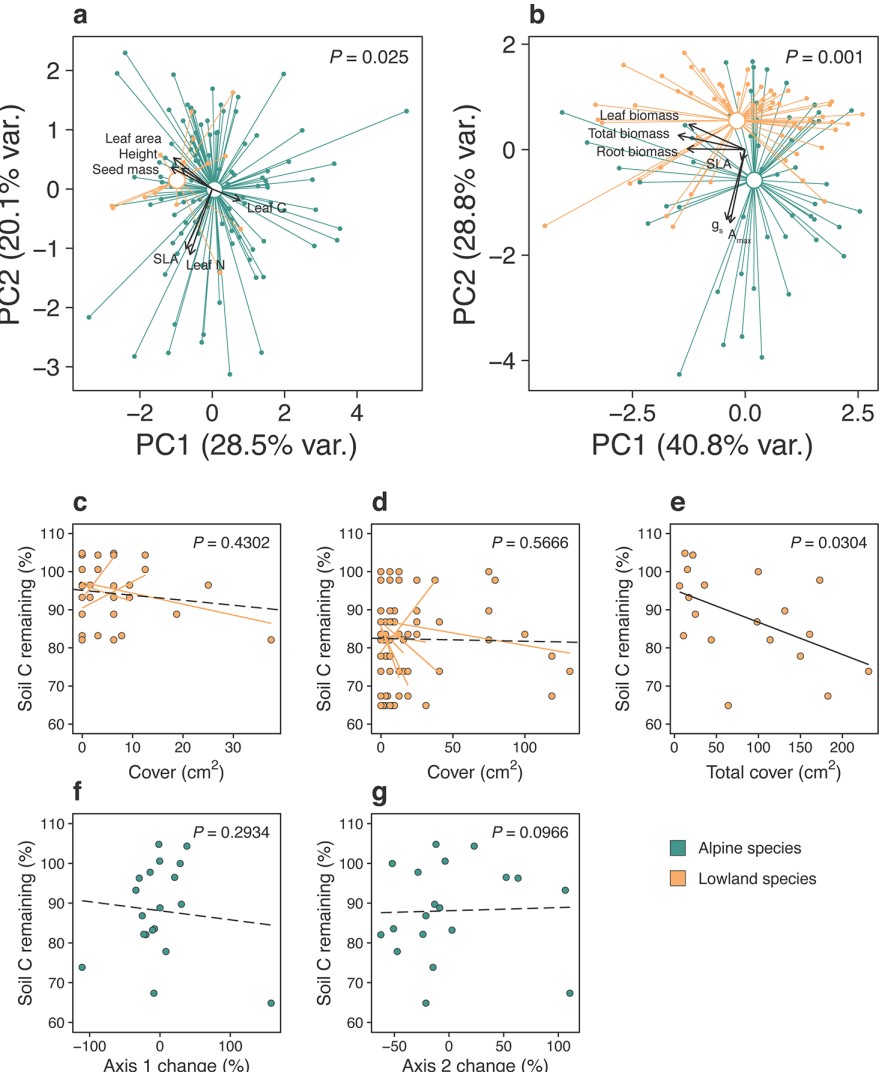

**Appendix 1—figure 3.** Alpine and lowland plant effects on soil carbon content. Trait separation of lowland (yellow) and alpine (green) plant species, displayed as PC1 and PC2 scores from PCAs containing (**a**) TRY data for species in field experiments ($N = 242$; $F_{1,240} = 3.98$, p = 0.003) or (**b**) measured data from a glasshouse experiment on selected species ($N = 109$; $F_{1,107} = 17.69$, p = 0.001). Arrows show loadings and p values refer to alpine–lowland comparison (PERMANOVAs). Relationships between soil carbon loss (% initial soil carbon content remaining) and cover (cm²) of each lowland plant species in the (**c**) western Alps ($N = 36$; LME: LR = 0.62, p = 0.4302) and (**d**) central Alps ($N = 80$; LME: LR = 0.33, p = 0.5666) experiments, (**e**) total lowland plant cover (cm²; $N = 19$; see Main text) and (**f, g**) alpine plant community composition (% change in NMDS axes; $N = 18$; linear models: NMDS #1: $F_{1,14} = 1.19$, p = 0.2934, NMDS #2: $F_{1,14} = 3.17$, p = 0.0966; region: $F_{1,14} = 1.29$, p = 0.2755). For (**c, d**), non-significant species-wise fit lines (yellow) are also displayed.

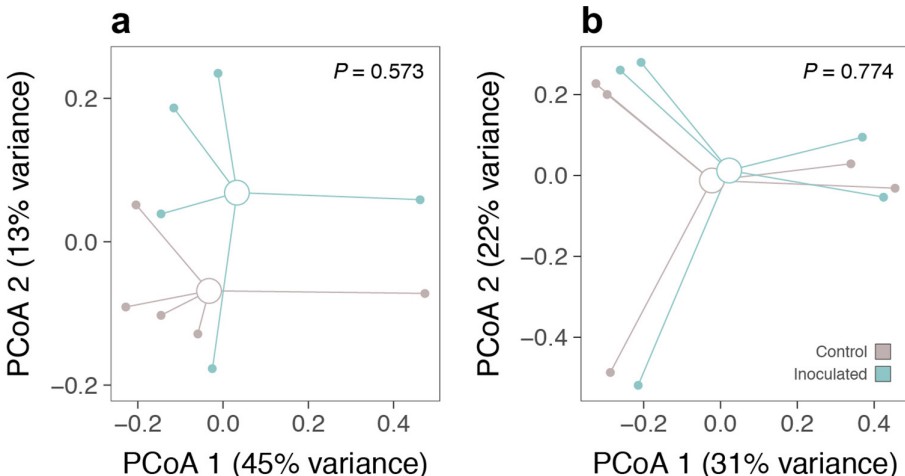

**Appendix 1—figure 4.** Soil inoculation effects on alpine soil community composition. Effects of low-elevation soil biota inoculation on (**a**) bacterial and (**b**) fungal community composition in alpine plots (*N* = 10 in both cases). Axes show the first two components of principal coordinates analyses (PCoA; Bray–Curtis distance) performed on the relative abundances of bacterial and fungal operational taxonomic units, respectively. p values are from PERMANOVAs testing for inoculation effects on the same distance matrices.

