## [Editor Report]

The authors transplanted alpine turfs from their cold environment to a lowland warm environment. They found that when lowland plants were inserted into these turfs under the thus simulated warming treatment they rapidly increased soil microbial decomposition of carbon stocks due to root exudates feeding the soil microbes. This finding is relevant because it suggests that global warming and shifts in plant species distributions may cause the release of soil-stored carbon into the atmosphere, thus further increasing warming.

---

## [Decision Letter]

**Decision letter after peer review:**

[Editors’ note: the authors submitted for reconsideration following the decision after peer review. What follows is the decision letter after the first round of review.]

Thank you for submitting the paper "Lowland plant migrations into alpine ecosystems amplify soil carbon loss under climate warming" for consideration by *eLife*. Your article has been reviewed by 4 peer reviewers, including Bernhard Schmid as the Reviewing Editor and Reviewer #1, and the evaluation has been overseen by a Senior Editor. The following individual involved in review of your submission has agreed to reveal their identity: Xin Jing (Reviewer #4).

Comments to the Authors:

You will see that the reviewers found your study highly innovative and well conducted. However, they also raise substantial concerns that lead us to decline the manuscript in its current form. If you feel you could incorporate the suggestions of the reviewers in a new manuscript, we would be happy to consider a new submission.

*Reviewer #1:*

The authors of this study carried out two carefully designed field and a glasshouse experiment simulating effects of rapid warming on soil carbon loss. They did this by transplanting alpine turfs from their cold environment to lowland warm environment. They found that when lowland plants were inserted into alpine turfs under these lowland climatic conditions (referred to as warming treatment combined with warm-adapted plant introduction) they rapidly increased soil microbial decomposition of carbon stocks due to root exudates feeding the microbes.

The question is how well this experimental setup mimics what would happen if lowland plants would be inserted into alpine turfs in situ (which have already experienced considerable warming over the past decades), perhaps with an additional warming treatment there. A further question is if alpine plants inserted in turfs at alpine climatic conditions would have a similar effect as lowland plants inserted in turfs at lowland climatic conditions.

I suggest that the authors consider these questions when they draw conclusions about the results from their experiments. It would also be interesting to discuss the relevance of sudden strong warming effects relative to slower warming, potentially allowing ecosystems to adjust via changes in genetic composition of species (i.e. evolution) or species composition of communities (i.e. community assembly).

Overall, I was very impressed by this study regarding experimental designs and analysis. However, as I read the three reviews of colleagues and the interpretations of the results I found that the authors in my view draw general conclusions that are not justified with this confidence.

Thus, I find some of the phrasing misleading, in particular already the shortcut to use warming for transplanting alpine turfs to lowland conditions. Thus, one of the other reviewers first thought you had transplanted lowland plants to alpine turfs (which also have warmed already over the past decades). You end paragraphs very nicely with suggestions or implications, but I think more parsimonious interpretations would often relate to the inability of alpine plants to have the same effects as lowland plants under the lowland climatic conditions because of lack of adaptation. For example, on lines 111-113 you could write "alpine plants transplanted to lowland climatic conditions reduce growth, root exudation and thus microbial activity and soil carbon loss". Or, on lines 133-135: "resident alpine plants cannot induce soil carbon loss from soil transferred to environmental conditions of lowland plants." Or, lines 151-152: "that the home temperature environment is more important than the home soil environment."

Regarding the statistical analysis I have difficulties to believe the extremely high significances of effects for lowland vs. alpine plants in the glasshouse experiment given that there were only n=3 replicate species for the two groups. I do believe that significances would remain if species identity is added as random-effect to the LME error model, but currently I did not find out if this was the case.

*Reviewer #2:*

The authors were trying to test whether the migration of lowland plants into alpine ecosystems affects the warming impact on soil carbon. To achieve this goal, the authors first did two field experiments (moving intact turf from high-elevation to low-elevation to simulate warming) in the Alps, and then did a greenhouse pot study to explore the potential mechanisms for the results observed in the field experiments.

The main strengths of this work are the combination of a field experiment (conducted at two sites) and a greenhouse pot experiment (to explore the detailed mechanisms). Moreover, a number of techniques were used to measure plant traits, soil DOM and microbial properties (e.g. CUE, growth) which help to find the potential mechanisms.

The main weaknesses of this work are below:

1) The two field experiments are very short-term (<1 year), but the results were that warming and/or warming+lowland plants led to very high amount of soil C loss (up to ~40%, Figure 1). I was shocked to see these results as many field warming studies have shown undetectable change in SOC even after years or decades. The authors did not provide a good explanation for this rapid and large change in SOC.

2) The greenhouse experiment was used to explore the potential reasons for the amplified loss of soil C in the field experiment. However, a key result was based on incubation of disturbed soils (8 g) and a two-pool modeling of the respiration data from the short-term incubation. This may not provide a good estimate of the true turnover rate of SOC under different plant species (even in the greenhouse condition). If rhizosphere priming was the proposed mechanism (as hinted by the authors), a better approach (such as 13C labeling) is needed to measure microbial respiration from intact soils (with plant/root presence).

3) Some details of the sampling or measurement are very crucial and affect the results/interpretations. For example, in the field experiment, the soil core was only 1-cm diameter. Considering the spatial heterogeneity of soil carbon in field plots, this small volume may not well represent the true soil condition. Moreover, in the field plots, did soil bulk density change after planting of lowland plants or warming? This will affect the measured SOC concentration (mg/g) even the SOC stock (g/m2) did not change.

The authors provide a lot of data from two well-replicated experiments, which is applaudable. However, I have concerns on some of the key results and conclusions as explained above. I would like the authors to further elaborate these concerns.

L32: what does this 52% mean? If the baseline is low (e.g. 2% loss in SOC relative to control), a 52% increase in the warming effect (i.e. 3% loss in SOC relative to control) is not a big deal. I think the percent change relative to the unwarmed control really matters here.

Figure 1: I was shocked to see these SOC data. Other data of C pools (e.g. AGB, BGB, MBC) may help interpret this.

*Reviewer #3:*

The authors investigated the effect of warming and herbaceous plant migration on soil carbon (C) content using an ecosystem monolith transplant experiment along an elevation gradient in the Swiss Alp mountains. They observed, approximately 1 year after the transplant, that warming alone had little effect on soil carbon content (monoliths transplanted to a lower elevation with higher temperature remained unchanged in C content) but that the presence of lowland (warm-adapted) herbaceous plants in combination with warming had a negative effect on soil C content. The authors then conducted a glasshouse experiment and used a series of field and laboratory measurements to explore potential mechanisms explaining the observed changes in soil C content in the field. They concluded that soil C losses under lowland plant migration were likely mediated via increased microbial activity and CO2 release from soil C decomposition.

The research questions are extremely relevant to our understanding of the feedback between soil C dynamics and climate warming and remain an unexplored part of this debate. Moreover, both field and laboratory experimental designs are robust, with all the relevant and necessary validation checks needed for transplant experiments; the laboratory techniques employed to measure the range of microbial and plant variables potentially explaining soil C dynamics are adequate and modern; and the statistical analyses are appropriate. These elements make the present data set very relevant and valuable. The manuscript is also very well and clearly written.

However, I have two major concerns, casting doubt respectively on the main field results and on the proposed explanatory mechanisms.

First, at no point is bulk density mentioned and it does not appear to have been measured. This is critical because changes in soil C concentration (which was measured and reported here, in mg C g-1 soil) does not necessarily indicate an actual change in the quantity of C present in the soil (C stock, in unit mass C per unit soil volume, or per unit surface area to a constant depth) if this is accompanied by a change in bulk density: if less C per unit mass of soil (lower C concentration) is concurrent with more mass of soil in a constant volume (higher bulk density), this could mean that no change in C stocks actually occurs (or that even an increase occurs). In the present study, it is possible that the presence of lowland plants increased bulk density as compared to only alpine plants, compensating the lower C concentration and resulting in no change in C stocks. This is perhaps not likely, but it is too critical an issue not to be quantified (or at the very least discussed).

Second, even assuming that no changes in bulk density occurred and that indeed soil C stocks decreased under warming combined with lowland plant migration, the interpretation of the results are, in my view, at least incomplete. Certainly, the results do not support the claim that soil C losses were mediated via increased microbial decomposition of soil C with the certainty suggested by the authors. Generally speaking, I see three issues with the interpretation:

– Very schematically, increased microbial respiration and soil C losses from decomposition is only one of two equally likely pathways potentially explaining soil C losses (the other being decreased C inputs to the soil from the plant community). The possibility that decreased soil C content was simply mediated by decreased inputs of C to the soil is hardly explored at all in the study (there is a quick mention of it (L155)), but differences in plant biomass are interpreted only for their correlations with microbial activity (L160-166), not as a component of the C balance. Plant traits are measured and analysed but not in a way that can be used to test the hypothesis of changing C inputs. The presence of "more productive traits" (L141) for the lowland plants does not directly relate to differences in the quantity of C inputs to the soil, nor is it interpreted in relation to inputs. Even the interpretation of changes in ecosystem respiration seem to omit the possibility of changes in plant respiration (L208): "depressed microbial respiration per unit of soil was also evident at the ecosystem scale in that warming accelerated total ecosystem respiration but its effect was dampened in plots containing lowland plants". This statement was made despite no significant differences in microbial respiration per unit soil in the field data, and disregards the possibility that the dampened effect in plots with lowland plants could be due to lower plant respiration.

– For the glasshouse experiment, I agree that the results indicate that (L115); "lowland plants accelerated microbial activity by increasing the quantity of root exudates", but not that (L112): "these findings together imply that lowland plants accelerate alpine soil C loss" because stimulating microbial activity is not per se an indicator of soil C loss. It is now well-known that the activity of microbes is not only a motor for soil C losses, but also a key mechanism leading to transformation of C inputs from plants that leads to the subsequent stabilisation of C in the soil. This is actually clearly stated further down in the manuscript when interpreting the field microbial data (L190). Furthermore, there is no direct evidence that the pots with lowland plants were losing more C than those without. Therefore, results from the glasshouse experiment could be interpreted differently: a larger fast cycling pool of soil C constituted of recently photosynthetically fixed exudates associated with higher microbial activity could well be interpreted as an early indicator of more C stabilisation, particularly since the absorbance index seems to indicate more microbially derived product in the DOC. It would have been great to measure microbial biomass C over time (as well as CUE, and mass specific growth and respiration), to see if higher respiratory activity was associated with higher biomass. The lack of differences in microbial biomass between the plant community treatments at the end of the 6 weeks does not show that the quantity of microbial biomass produced over the whole incubation period remained constant. In a word, more respiration of a larger fast cycling pool is not an indicator of future soil C loss (in the presence of plants).

– The interpretation of the microbial variables measured in the field line up better with current conceptualisations of the role of microbes in C cycling (but overall interpretation still lacks consideration for plant C inputs). However, interpreting those data measured once 1 year after the transplant to explain the changes that happened gradually over this whole year is a risky and difficult exercise. How do we know that CUE, Rmass, Gmass etc… measured then represent what they were a day, a week, a month before? There is an attempt to deal with this timing issue by comparison with the glasshouse experiment, but only Cmic and Rmass can really be compared and it only very partially fills in the gap in time. Besides, the interpretation of this comparison can be questioned: in the glasshouse, Rmass was higher for the lowland plant pots (as compared to alpine plant at constant temperature) but actually remained constant between the comparable treatments W and WL in the field (Figure 2m). The results from the field, therefore, do not "support observations from the glasshouse experiment" in this context (L197) and neither do they "confirm (…) that this persists for at least one season" (L199). Finally, the thinking around the pulsed nature of C losses seems misplaced because there are no evidence that soil C losses had stopped after a year in the field (no measurements of soil C content are presented after that year).

In my view, it is critical that the authors address the bulk density issue. Ideally, bulk density data have been collected and the soil C content results (Figure 1) can be converted to C stocks. If bulk density data have not been collected, the effect of lowland plant migration at similar densities than in the transplant experiment on bulk density should be tested and confirmed to be neutral.

That being done, dC/dt = inputs – decomposition.

Nearly every aspect of the dataset should be re-interpreted in terms of a shifting balance between inputs and decomposition, rather than simply in terms of more or less microbial respiration. It could mean having to conduct additional analyses on the plant traits. To put it bluntly, it feels like the microbial data were interpreted as if collected from a soil incubation experiment with no plants (this is not totally fair of course because plant traits were measured, but not really interpreted in a way that is helpful to conclude on the input/decomposition balance).

In my view, the manuscript needs nearly a complete re-write, but I reiterate that this is a novel, valuable and properly (but perhaps not completely) analysed (statistically) data set.

*Reviewer #4:*

This manuscript took alpine grasslands as a model system and investigated whether lowland herbaceous plants contributed to the short-term dynamics of soil carbon under the context of climate warming. The authors find that warming individually does not render significant changes in alpine soil carbon, but corporately causes ~52% of carbon loss with lowland herbaceous plants in two short periods of field experiments. They further show that alpine soil carbon loss is likely mediated by lowland herbaceous plants through root exudation, soil microbial respiration, and CO2 release. This work adds in an interesting way to the ongoing debate on whether a positive climate feedback will be mediated by plant uphill range expansion in alpine grasslands, where climate warming may lead to a rapid loss of soil carbon.

The claims of this manuscript are well supported, but some aspects of background information in the studied alpine systems and field experiment design need to be clarified.

1) There is an extremely high level of carbon stored in the alpine soils (Figure 1). Climate warming will certainly lead to a great loss of soil carbon in the study systems that could contribute to the positive climate feedback. However, it is unclear for me how the effects of climate warming on soil carbon are relevant to the ongoing climate change in the studied alpine grasslands. It is therefore reasonable to provide more background information about ongoing climate change, and whether the simulated climate warming (i.e., 2.8 oC in central alps and 5.3 oC in western alps, Line 328-329) is realized as real-world climate change in the local systems. In addition, it seems that the manuscript aims to address a question that is of global concern, but my concern is about how the findings could be generalized to other regions.

2) I understand that the manuscript considers elevation as a natural gradient of climate change, which makes it possible to compare soil carbon dynamics in lowlands with alpine grasslands under climate warming. I also understand that the authors have done everything they can to control for the disturbances caused by transplanting that has been well justified by the supplementary data (e.g., Figure S6). However, it is unclear how the authors controlled for the influences of other factors given there are huge differences between lowlands and alpine grasslands, such as differences in wind, solar radiation, humidity, and the length of growing season.

3) It is generally known that different species respond to climate warming differently. Some species may be sensitive to climate warming and have traits aiding to dispersion that could expand their living ranges to some degree, while others may adjust themselves to adapt to climate warming and may not migrate to alpine systems. It is therefore cautious to assume that all the lowland species have the same dispersal ability. In other words, it is unclear how lowland plant species are selected for the field transplanting experiment (Line 284-290). Do all the lowland plant species selected have the potential to migrate to alpine systems?

4) The authors acknowledge that "we did not perform a reverse transplantation (that is, from low to high elevation), so we cannot entirely rule out the possibility that transplantation of any community to any new environment could yield a loss of soil carbon" (Line 318-320). When I read the title "lowland plant migrations into alpine grasslands …", I thought lowland plant species that were transplanted from low to high elevation. In fact, it is just the opposite to my thoughts. Without performing a reverse transplantation experiment, I am not sure the conclusion will stand that "lowland plant migrations into alpine grasslands amplify soil carbon loss under climate warming". In addition, it is unclear whether lowland plant effects stand alone or depend on climate warming based on the results in Figure 1 that lowland plant treatment is missing, and it is impossible to test the interactions between lowland plant and climate warming.

---

## [Author Response]

[Editors’ note: the authors resubmitted a revised version of the paper for consideration. What follows is the authors’ response to the first round of review.]

Reviewer #1:The authors of this study carried out two carefully designed field and a glasshouse experiment simulating effects of rapid warming on soil carbon loss. They did this by transplanting alpine turfs from their cold environment to lowland warm environment. They found that when lowland plants were inserted into alpine turfs under these lowland climatic conditions (referred to as warming treatment combined with warm-adapted plant introduction) they rapidly increased soil microbial decomposition of carbon stocks due to root exudates feeding the microbes.The question is how well this experimental setup mimics what would happen if lowland plants would be inserted into alpine turfs in situ (which have already experienced considerable warming over the past decades), perhaps with an additional warming treatment there.

The Reviewer alludes to two pertinent points here. The Reviewer’s first point considers whether lowland plants would function similarly (and, by extension, have the same effect on the soil system) if moved from the warmer lowland site to the cooler alpine site. This is a fascinating question in its own right, in that it raises questions about how migrations of non-adapted genotypes far beyond range edges (e.g. *via* human activity) impact recipient ecosystems. However, although we agree that alpine ecosystems have warmed considerably in recent decades, we cannot be confident that the high elevation sites in our study are already within the climate niche of the lowland focal species. As such, to address our research questions in situ at the high sites would have required additional warming treatments, which come with their own set of disadvantages (see our second point to this comment, below). We also refer the Reviewer to specific questions about adaptation below (see R6), although we see that we were not careful enough about the rationale for our design in the previous version of the manuscript. We have therefore added a clarifying sentence to the Main Text as follows:

L101: “In short, the experiments used here examined how the arrival of warm-adapted lowland plants influences alpine ecosystems in a warmed climate matching lowland site conditions (i.e. turf transplantation to low elevation plus lowland plant addition) relative to warming-only (i.e. turf transplantation to low elevation) or control (i.e. turf transplantation within high elevation) scenarios.”

Second, the Reviewer implicitly raises a point about whether our chosen approach of simulating warming plus lowland plant arrival (i.e. transplantation plus addition of lowland plants) is the most appropriate, specifically by suggesting an alternative option of adding lowland plants to (possibly experimentally-warmed) alpine turfs at the high elevation origin site. Here, it was essential to create a climate scenario in which lowland plants would survive and operate within their climatic niche (i.e. relative to their home conditions) once planted into alpine turfs, rather than perform suboptimally (e.g. be in a potentially inferior competitive position) or be unable to persist at all. The most parsimonious and reliable way to ensure this was to transplant alpine turfs to a site with a lowland temperature regime, with transplantations also being shown to outperform other methods when novel species interactions are involved (Yang *et al.* 2018). Most importantly, it was crucial to select a method that warmed the entire plant-soil system rather than only the air (e.g. open-top chambers, IR lamps; Marion *et al.* 1997; Aronson *et al.* 2009) or soil (e.g. heating cables; Hanson *et al.* 2017), and did so realistically throughout the year regardless of the weather (e.g. open-top chambers only work on sunny days in the summer; Marion *et al.* 1997) or a power supply (e.g. IR lamps, heating cables). Transplantation remains the only way to achieve this (Hannah 2022; Shaver *et al.* 2000). We now clarify our logic in the manuscript as follows:

L91: “Elevation-based transplant experiments are powerful tools for assessing climate warming effects on ecosystems because they expose plots to a real-world future temperature regime with natural diurnal and seasonal cycles while also warming both aboveground and belowground subsystems^25–27^. This is especially true if they include rigorous disturbance controls (here, see Methods) and are performed in multiple locations where the common change from high to low elevation is temperature (here, warming of 2.8 ºC in the central Alps and 5.3 ºC in the western Alps). While factors other than temperature can co-vary with elevation^17^, such factors either do not vary consistently with elevation among experiments (e.g. precipitation, wind), are not expected to strongly influence plant performance (e.g. UV radiation) or in any case form part of a realistic climate warming scenario (e.g. growing-season length, snow cover)^17,27,28^.”

A further question is if alpine plants inserted in turfs at alpine climatic conditions would have a similar effect as lowland plants inserted in turfs at lowland climatic conditions.

We interpret “turfs” to mean “lowland turfs” here, since we did insert lowland plants into alpine turfs under lowland climatic conditions (i.e. the WL treatment). We found that adding alpine plants to alpine turfs in alpine climatic conditions (i.e. planting disturbance control, see Methods) had no effect on alpine soil carbon content (L458). By extension, we would expect that adding lowland plants to lowland turfs in lowland climatic conditions would have no effect on lowland soil carbon content. While not explicitly tested, including this treatment would not change our finding that adding lowland plants to alpine turfs causes a reduction in soil carbon content relative to adding alpine plants to alpine turfs. Given this, we have left the text as is, but are happy to revisit this issue based on further discussion with the Reviewer/Editor.

I suggest that the authors consider these questions when they draw conclusions about the results from their experiments. It would also be interesting to discuss the relevance of sudden strong warming effects relative to slower warming, potentially allowing ecosystems to adjust via changes in genetic composition of species (i.e. evolution) or species composition of communities (i.e. community assembly).

Thank you for this excellent suggestion. We absolutely agree that anything short of a decadal experiment is unable to detect the role of longer-term evolutionary or community processes on soil carbon dynamics. While this doesn’t eliminate the need for experiments that consider shorter timescales, it is important to explicitly state this limitation. As suggested, we have added a sentence discussing this possibility in the concluding paragraph:

L387: “While our findings demonstrate that lowland plants affect the rate of soil carbon release in the short term, short-term experiments, such as ours, cannot resolve whether lowland plants will also affect the total amount of soil carbon lost in the long term. This includes whether processes such as genetic adaptation (in both alpine and lowland plants)^12^ or community change^44,45^ will moderate soil carbon responses to gradual or sustained warming.”

We also agree that it is extremely challenging to undertake warming experiments that do not initially “shock” the system through a sudden change in temperature. Having said this, alpine ecosystems are adapted to rapid within- and between-season temperature changes, making such shocks less relevant here.

Overall, I was very impressed by this study regarding experimental designs and analysis. However, as I read the three reviews of colleagues and the interpretations of the results I found that the authors in my view draw general conclusions that are not justified with this confidence.Thus, I find some of the phrasing misleading, in particular already the shortcut to use warming for transplanting alpine turfs to lowland conditions. Thus, one of the other reviewers first thought you had transplanted lowland plants to alpine turfs (which also have warmed already over the past decades).

We agree with the Reviewer that there was a leap in logic in the original submission between transplantation and warming. We have resolved this in two ways. First, we now explicitly state the rationale behind using transplant experiments in the first paragraph of the Results:

L91: “Elevation-based transplant experiments are powerful tools for assessing climate warming effects on ecosystems because they expose plots to a real-world future temperature regime with natural diurnal and seasonal cycles while also warming both aboveground and belowground subsystems^25–27^. This is especially true if they include rigorous disturbance controls (here, see Methods) and are performed in multiple locations where the common change from high to low elevation is temperature (here, warming of 2.8 ºC in the central Alps and 5.3 ºC in the western Alps). While factors other than temperature can co-vary with elevation^17^, such factors either do not vary consistently with elevation among experiments (e.g. precipitation, wind), are not expected to strongly influence plant performance (e.g. UV radiation) or in any case form part of a realistic climate warming scenario (e.g. growing-season length, snow cover)^17,27,28^. In short, the experiments used here examined how the arrival of warm-adapted lowland plants influences alpine ecosystems in a warmed climate matching lowland site conditions (i.e. turf transplantation to low elevation plus lowland plant addition) relative to warming-only (i.e. turf transplantation to low elevation) or control (i.e. turf transplantation within high elevation) scenarios.”

Second, we have altered the language used to describe results throughout the manuscript to qualify that warming was simulated through transplantation. Specifically, we no longer use transplantation as a shortcut for warming, but describe the results in terms of the experimental manipulations rather than in terms of the scenarios they are intended to simulate. Correspondingly, our conclusions are now interpreted as implications of the experimental treatments. We now see the potential for our previous formulations to be misleading, and have therefore ensured that all conclusions are phrased in a way that is fully justified by the results. For instance:

L77: “Figure 1 | Warming and lowland plant effects on alpine soil carbon content in the field experiment. Mean ± SE soil carbon content (mg C g^-1^ dry mass; i.e. mass-based per-mil) in alpine turfs transplanted to low elevation (warming, W; light grey), transplanted plus planted with lowland plants (warming plus lowland plant arrival, WL; dark grey) or replanted at high elevation (control, C; white). Data are displayed for two experiments in the western (left) and central (right) Alps, with letters indicating treatment differences (LMEs; N = 58).”

L107: “We found that transplantation from high to low elevation (i.e. warming-only) caused a small decline in alpine soil carbon content, but in both regions this effect was stronger and only became statistically significant when lowland plants were also planted-in (Figure 1; Table S1; *P* < 0.001). Specifically, considering field data from both experiments together, the addition of lowland plants to transplanted alpine turfs increased warming-induced soil carbon loss from 4.1 ± 0.23 mg g^-1^ per ºC to 6.3 mg g^-1^ per ºC, which is a 52% ± 31% (mean ± 95% confidence intervals) increase relative to warming alone (Supplementary Figure S3; *P* = 0.001).”

L254: “Based on these observations, we hypothesise that, at least in our experiments, the establishment of warm-adapted lowland plants in warmed alpine ecosystems introduces novel traits into the community that alter plant community functioning and thus carbon cycle processes, with effects that intensify as lowland plants become more abundant in the community^31,32^. An alternative explanation is that, trait differences aside, the ability of alpine plants to facilitate soil carbon loss is suppressed when growing in warmer climates to which they are not adapted.”

L303: “… Effects of transplantation to low elevation (warming, W), transplantation plus lowland plants (warming plus lowlands, WL) or replantation at high elevation (control, C) on alpine ecosystem carbon dioxide fluxes (mean ± SE) during one season.”

L311: “… Effects of transplantation to low elevation (warming, W), transplantation plus lowland plants (warming plus lowlands, WL) or replantation at high elevation (control, C) on alpine soil pools and processes (means ± SEs) after one season.”

L325: “We found that biomass-specific rates of microbial respiration in the field were greater following transplantation to the warmer site in the presence, but not absence, of lowland plants (Figure 6a; Table S1; *P* = 0.038). By contrast, biomass-specific rates of microbial growth were slower following transplantation but remained unaffected by lowland plant presence (Figure 6b; Table S1; *P* = 0.016).”

L338: “However, in the field we found that lowland plants caused a reduction of microbial biomass carbon in alpine communities transplanted to the warmer site (Figure 6c; Table S1; *P* < 0.001).”

L350: “In short, we propose based on these relationships that the decrease in soil carbon content observed in warmer transplanted plots containing lowland plants had both started and stopped within one season.”

L408: “The experiments simulate effects of climate warming on alpine ecosystems *via* transplantation from high to low elevation, including the arrival of lowland plants due to warming induced uphill plant migrations (Supplementary Figure S1).”

L442: “Overall, the experimental design in both regions yielded three treatments (n = 10 for each) arranged in a block design: transplantation from high to low elevation (i.e. warming-only, W); transplantation plus low elevation plants (i.e. warming plus lowland plant establishment, WL); and replantation at the high elevation origin site (i.e. control, C).”

You end paragraphs very nicely with suggestions or implications, but I think more parsimonious interpretations would often relate to the inability of alpine plants to have the same effects as lowland plants under the lowland climatic conditions because of lack of adaptation. For example, on lines 111-113 you could write "alpine plants transplanted to lowland climatic conditions reduce growth, root exudation and thus microbial activity and soil carbon loss". Or, on lines 133-135: "resident alpine plants cannot induce soil carbon loss from soil transferred to environmental conditions of lowland plants." Or, lines 151-152: "that the home temperature environment is more important than the home soil environment."

We thank the Reviewer for this important and thought-provoking comment. On reflection, we agree that the experimental design precludes us from identifying whether lowland plants affect warmed alpine soils because of their traits, *per se*, or because their performance is conditional on the warmer low site conditions. While this distinction does not influence the main findings of the study (i.e. that the arrival of warm-adapted lowland plants in warming alpine ecosystems facilitates soil carbon loss), it is a subtle but still important qualifier that was missing from the original submission. We have rectified this in the revised manuscript in four ways.

First, we have clarified that the experimental design explicitly tests the scenario whereby lowland plants arrive in alpine ecosystems experiencing a warmer climate akin to their home conditions:

L101: “In short, the experiments used here examined how the arrival of warm-adapted lowland plants influences alpine ecosystems in a warmed climate matching lowland site conditions (i.e. turf transplantation to low elevation plus lowland plant addition) relative to warming-only (i.e. turf transplantation to low elevation) or control (i.e. turf transplantation within high elevation) scenarios.”

Second, we agree with the Reviewer that there are two directions with which to interpret the findings. On the one hand, lowland plants facilitate soil carbon loss under the warmed conditions to which they are adapted. On the other hand, alpine plants cannot induce soil carbon loss to the same extent as lowland plants under warmer conditions due to a lack of adaptation. Having said this, in our opinion alpine plants must remain the baseline reference point for interpretation, since the focus is on how alpine ecosystems change once warming occurs and lowland plants arrive. So, while we absolutely agree that adaptation of lowland plants to the lowland site may play an important role in the response observed, lowland plants still represent the “altered state”. We have thus decided against reversing the direction of comparison (i.e. that alpine plants are suppressed relative to lowland plants at lowland conditions), although we now present this issue as an alternative perspective for consideration. Specifically:

L254: “Based on these observations, we hypothesise that, at least in our experiments, the establishment of warm-adapted lowland plants in warmed alpine ecosystems introduces novel traits into the community that alter plant community functioning and thus carbon cycle processes, with effects that intensify as lowland plants become more abundant in the community^31,32^. An alternative explanation is that, trait differences aside, the ability of alpine plants to facilitate soil carbon loss is suppressed when growing in warmer climates to which they are not adapted. While either explanation could ultimately accelerate soil carbon loss from warming alpine ecosystems following the arrival of lowland plants, future studies are needed to fully unravel the environmental dependence of plant trait effects on soil ecosystem processes.”

Third, we have nuanced our language throughout the manuscript to make it clear that the effect of lowland plants on the alpine turfs is conditional on the warmer lowland site conditions. That is to say, we avoid implying that alpine and lowland plants have intrinsically different effects on the soil system (even though the functional trait literature would support such an interpretation), and rather emphasise that the differences we observe are conditional on the environmental conditions at the site (which is what our experimental design allows us to conclude). For instance:

L67: “We first used two whole-community transplant experiments in different alpine regions (western/central Swiss Alps) to establish the effects of warming plus lowland plant arrival on alpine soil carbon content.”

L89: “A portion of the alpine turfs at low elevation were planted with local lowland plant species to simulate the arrival of warm-adapted lowland plants in the warmed ecosystem, with the remaining portion being subjected to a planting disturbance control.”

L115: “Nevertheless, these findings show that once warm-adapted lowland plants establish in warming alpine communities, they facilitate warming effects on soil carbon loss on a per gram basis.”

Fourth, with reference to L111-L113 in the original manuscript, the findings in question relate to the glasshouse experiment, which was undertaken in standardised conditions without a warming treatment *per se*. We see that this was not clear in the Main Text of the original submission, so we have added a sentence to clarify this as follows:

L127: “The glasshouse experiment was not a direct simulation of field conditions, in that plants from lowland and alpine communities were grown separately at a constant temperature and humidity (Methods). Instead, the glasshouse experiment allowed us to isolate how plants adapted to lowland versus alpine climates differentially affect the alpine soil system.”

We have also nuanced the concluding sentence of this paragraph as follows:

L181: “These findings support the hypothesis that lowland plants have the capacity to increase soil carbon outputs relative to alpine plants by stimulating soil microbial respiration and associated CO_2_ release.”

With reference to the remaining examples given by the Reviewer, we have retained the original direction of comparison (see above) but added nuance to allow for adaptation to be responsible. Specifically:

L215: “While further directed studies are required to resolve whether root exudates are truly involved, our findings collectively suggest that lowland plants have the capacity to increase total root exudation into alpine soil relative to resident alpine plants.”

L256: “Based on these observations, we hypothesise that, at least in our experiments, the establishment of warm-adapted lowland plants in warmed alpine ecosystems introduces novel traits into the community that alter plant community functioning and thus carbon cycle processes, with effects that intensify as lowland plants become more abundant in the community^31,32^.”

Regarding the statistical analysis I have difficulties to believe the extremely high significances of effects for lowland vs. alpine plants in the glasshouse experiment given that there were only n=3 replicate species for the two groups. I do believe that significances would remain if species identity is added as random-effect to the LME error model, but currently I did not find out if this was the case.

We apologise for the confusion here. The glasshouse experimental design involved two planted treatments (not including the bare soil treatment) – one planted with lowland species and another planted with alpine species. We used six species from each alpine and lowland group, but every species was planted into every pot within its respective group. As such, there was no within treatment variation in species composition, with species composition only varying between alpine and lowland plant treatments. In other words, each of the 10 replicates for the lowland or alpine plant treatments were identical in terms of species composition – and varied only in their position in the greenhouse. We accounted for such a blocked design using linear mixed effects models. We have now clarified this in the revised manuscript:

L587: “Vegetated treatments contained two individuals of all lowland or alpine species planted in a circular pattern such they were never adjacent to conspecifics or plants of the same functional type (i.e. grass, herb, legume), making replicates within treatments identical in species composition and neighbourhoods.”

L718: “… we used the glasshouse experiment to test for lowland versus alpine plant treatment effects on soil microbial biomass carbon, the size (*C_l_*) and decay rate (*k_l_*) of the first modelled soil carbon pool and all variables relating to soil DOM quality using LMEs including block as a random intercept term.”

L725: “Finally, we explored how plant physiology in lowland and alpine treatments at the pot level affected initial rates of soil microbial respiration.”

Reviewer #2:The authors were trying to test whether the migration of lowland plants into alpine ecosystems affects the warming impact on soil carbon. To achieve this goal, the authors first did two field experiments (moving intact turf from high-elevation to low-elevation to simulate warming) in the Alps, and then did a greenhouse pot study to explore the potential mechanisms for the results observed in the field experiments.The main strengths of this work are the combination of a field experiment (conducted at two sites) and a greenhouse pot experiment (to explore the detailed mechanisms). Moreover, a number of techniques were used to measure plant traits, soil DOM and microbial properties (e.g. CUE, growth) which help to find the potential mechanisms.

We thank the Reviewer for this positive comment.

The main weaknesses of this work are below:1) The two field experiments are very short-term (<1 year), but the results were that warming and/or warming+lowland plants led to very high amount of soil C loss (up to ~40%, Figure 1). I was shocked to see these results as many field warming studies have shown undetectable change in SOC even after years or decades. The authors did not provide a good explanation for this rapid and large change in SOC.

We apologise for the confusion. We’re unsure where “up to ~40%” comes from here, so we have taken the Reviewer’s later suggestion of changing the annotation on Figure 1 to contrast C versus WL treatments (Western Alps = 25.6 ± 7.2 mg g^-1^; Central Alps = 25.3 ± 8.6 mg g^-1^) rather than W versus WL treatments.

With regards to the magnitude of soil carbon loss observed, we express soil carbon content in mg g^-1^ (i.e. mass-based per-mil), not cg g^-1^ (i.e. mass-based percent). This is so that we could use percent changes in the text to highlight the numeric magnitude of differences between treatments without confusing them with mass-based percent soil carbon – although we appreciate that this also caused confusion (see R15). To clarify, converting the above C versus WL treatment contrasts from mg g^-1^ to mass-based percent yields 2.56% ± 0.72% for the Western Alps experiment and 2.53% ± 0.86% for the Central Alps experiment. While it is striking that the WL treatments lost ~2.5% (~25 mg g^-1^) soil carbon in one year, such a loss is not extraordinary. To avoid future confusion, we have clarified the units in the Figure 1 caption as follows:

L77: “Mean ± SE soil carbon content (mg C g^-1^ dry mass; i.e. mass-based per-mil) in alpine turfs transplanted to low elevation (warming, W; light grey), transplanted plus planted with lowland plants (warming plus lowland plant arrival, WL; dark grey) or replanted at high elevation (control, C; white). Data are displayed for two experiments in the western (left) and central (right) Alps, with letters indicating treatment differences (LMEs; N = 58).”

2) The greenhouse experiment was used to explore the potential reasons for the amplified loss of soil C in the field experiment. However, a key result was based on incubation of disturbed soils (8 g) and a two-pool modeling of the respiration data from the short-term incubation. This may not provide a good estimate of the true turnover rate of SOC under different plant species (even in the greenhouse condition). If rhizosphere priming was the proposed mechanism (as hinted by the authors), a better approach (such as 13C labeling) is needed to measure microbial respiration from intact soils (with plant/root presence).

We agree with the Reviewer that using an approach such as ^13^C-labelling would have provided more direct evidence that lowland plants cause a rhizosphere priming effect. However, although some of our evidence comes from disturbed soils (i.e. microbial respiration), some (i.e. soil pore water) also comes from intact pots prior to harvest and we now also include another line of evidence from plant root biomass (see R18). In short, we draw on multiple lines of evidence suggesting that root exudates were involved, and note that Reviewer #3 thought our approach and interpretation on this aspect of the study was robust.

Having said this, we acknowledge that we were too confident in our interpretation here, so we have added caveats to the text as follows:

L207: “While not directly measured here, a nine-day decay period corresponds to the time expected for newly photosynthesised CO_2_ to be released through root exudation and respired by soil microbes^23,24^, suggesting that this carbon pool was mostly root exudates.”

L215: “While further directed studies are required to resolve whether root exudates are truly involved, our findings collectively suggest that lowland plants have the capacity to increase total root exudation into alpine soil relative to resident alpine plants.”

3) Some details of the sampling or measurement are very crucial and affect the results/interpretations. For example, in the field experiment, the soil core was only 1-cm diameter. Considering the spatial heterogeneity of soil carbon in field plots, this small volume may not well represent the true soil condition. Moreover, in the field plots, did soil bulk density change after planting of lowland plants or warming? This will affect the measured SOC concentration (mg/g) even the SOC stock (g/m2) did not change.

We agree with the Reviewer that taking a single soil core of 1 cm diameter in each plot would not have been robust. We did not do this. While we used 1 cm diameter cores to minimise disturbance, we took three cores per plot to account for within-plot heterogeneity and combined them into a composite sample. This is stated in the Methods as follows:

L523: “In each plot, we created a composite sample from three cores (ø = 1 cm, approx. *d* = 7 cm) no closer than 7 cm from a planted individual and from the same quarter of the plot used for ecosystem respiration measurements (see below; Supplementary Figure S1).”

We also agree that bulk density measurements were an important omission in the initial submission. We note that this point was fleshed out by Reviewer #3, below, so we refer the Reviewer to our response to that comment for further details.

The authors provide a lot of data from two well-replicated experiments, which is applaudable. However, I have concerns on some of the key results and conclusions as explained above. I would like the authors to further elaborate these concerns.

We hope that our responses to the Reviewer’s detailed comments address these concerns.

L32: what does this 52% mean? If the baseline is low (e.g. 2% loss in SOC relative to control), a 52% increase in the warming effect (i.e. 3% loss in SOC relative to control) is not a big deal. I think the percent change relative to the unwarmed control really matters here.

We refer the Reviewer to our response to their previous comment which we hope has resolved this issue.

Figure 1: I was shocked to see these SOC data. Other data of C pools (e.g. AGB, BGB, MBC) may help interpret this.

We apologise for this. We believe that this comment is an extension to the Reviewer’s previous comments about the magnitude of soil carbon lost from the system. Either way, in response to comments from Reviewer #3 we now present new data on ecosystem carbon dioxide fluxes (Figure 5) and draw further attention to field soil pools and processes (Figure 6), which we hope resolves this issue.

Reviewer #3:The authors investigated the effect of warming and herbaceous plant migration on soil carbon (C) content using an ecosystem monolith transplant experiment along an elevation gradient in the Swiss Alp mountains. They observed, approximately 1 year after the transplant, that warming alone had little effect on soil carbon content (monoliths transplanted to a lower elevation with higher temperature remained unchanged in C content) but that the presence of lowland (warm-adapted) herbaceous plants in combination with warming had a negative effect on soil C content. The authors then conducted a glasshouse experiment and used a series of field and laboratory measurements to explore potential mechanisms explaining the observed changes in soil C content in the field. They concluded that soil C losses under lowland plant migration were likely mediated via increased microbial activity and CO2 release from soil C decomposition.The research questions are extremely relevant to our understanding of the feedback between soil C dynamics and climate warming and remain an unexplored part of this debate. Moreover, both field and laboratory experimental designs are robust, with all the relevant and necessary validation checks needed for transplant experiments; the laboratory techniques employed to measure the range of microbial and plant variables potentially explaining soil C dynamics are adequate and modern; and the statistical analyses are appropriate. These elements make the present data set very relevant and valuable. The manuscript is also very well and clearly written.

We thank the Reviewer and are delighted that they think the study is extremely relevant, novel, experimentally robust, cutting-edge and valuable.

However, I have two major concerns, casting doubt respectively on the main field results and on the proposed explanatory mechanisms.First, at no point is bulk density mentioned and it does not appear to have been measured. This is critical because changes in soil C concentration (which was measured and reported here, in mg C g-1 soil) does not necessarily indicate an actual change in the quantity of C present in the soil (C stock, in unit mass C per unit soil volume, or per unit surface area to a constant depth) if this is accompanied by a change in bulk density: if less C per unit mass of soil (lower C concentration) is concurrent with more mass of soil in a constant volume (higher bulk density), this could mean that no change in C stocks actually occurs (or that even an increase occurs). In the present study, it is possible that the presence of lowland plants increased bulk density as compared to only alpine plants, compensating the lower C concentration and resulting in no change in C stocks. This is perhaps not likely, but it is too critical an issue not to be quantified (or at the very least discussed).

This is an excellent point, and one also raised by Reviewer #2. To clarify, we initially decided against measuring bulk density because it is destructive and the experiments were still being used for other studies. Having said this, we agree with the Reviewer that more consideration of soil bulk density was needed, so we have rectified this in three ways. First, although the western Alps experiment has now been taken down, to address this comment we took new soil cores to measure bulk density in the central Alps experiment in 2021 to indirectly confirm that no changes occurred in the presence versus absence of lowland plants. They did not, and we now include these data in the Methods as follows:

L539: “It was not possible to take widespread measurements of soil bulk density due to the destructive sampling required while other studies were underway (e.g. ref ^28^). Instead, we took additional soil cores (ø = 5 cm, *d* = 5 cm) from the central Alps experiment in 2021 once other studies were complete to indirectly explore whether lowland plant effects on soil carbon content in warmed alpine plots could have occurred due to changes in soil bulk density. We found that although transplantation to the warmer site increased alpine soil bulk density (LR = 7.18, *P* = 0.028, Tukey: *P* < 0.05), lowland plants had no effect (Tukey: *P* = 0.999). It is not possible to make direct inferences about the soil carbon stock using measurements made on different soil cores four years apart. Nevertheless, these results make it unlikely that lowland plant effects on soil carbon content in warmed alpine plots occurred simply due to a change in soil bulk density.”

Second, in the Main Text we now caution readers against translating soil carbon content changes to soil carbon stock in absence of coupled measurements of soil bulk density as follows:

L113: “We caution against equating changes to soil carbon content with changes to soil carbon stock in the absence of coupled measurements of soil bulk density (Methods). Nevertheless, these findings show that once warm-adapted lowland plants establish in warming alpine communities, they facilitate warming effects on soil carbon loss on a per gram basis.”

Finally, we have altered the language throughout the manuscript (including the title) to make it clearer that we focussed on soil carbon content/concentration – not stock.

Second, even assuming that no changes in bulk density occurred and that indeed soil C stocks decreased under warming combined with lowland plant migration, the interpretation of the results are, in my view, at least incomplete. Certainly, the results do not support the claim that soil C losses were mediated via increased microbial decomposition of soil C with the certainty suggested by the authors. Generally speaking, I see three issues with the interpretation:– Very schematically, increased microbial respiration and soil C losses from decomposition is only one of two equally likely pathways potentially explaining soil C losses (the other being decreased C inputs to the soil from the plant community). The possibility that decreased soil C content was simply mediated by decreased inputs of C to the soil is hardly explored at all in the study (there is a quick mention of it (L155)), but differences in plant biomass are interpreted only for their correlations with microbial activity (L160-166), not as a component of the C balance. Plant traits are measured and analysed but not in a way that can be used to test the hypothesis of changing C inputs. The presence of "more productive traits" (L141) for the lowland plants does not directly relate to differences in the quantity of C inputs to the soil, nor is it interpreted in relation to inputs. Even the interpretation of changes in ecosystem respiration seem to omit the possibility of changes in plant respiration (L208): "depressed microbial respiration per unit of soil was also evident at the ecosystem scale in that warming accelerated total ecosystem respiration but its effect was dampened in plots containing lowland plants". This statement was made despite no significant differences in microbial respiration per unit soil in the field data, and disregards the possibility that the dampened effect in plots with lowland plants could be due to lower plant respiration.

This is an excellent point. We have performed new analyses of the plant trait/biomass data from the field experiment, included additional measurements/analyses of NEE and GPP from the field experiments (originally omitted due to space, which was a mistake!) and have rewritten all relevant sections in the manuscript to change the focus to a shifting balance between soil carbon inputs and outputs. We refer the Reviewer particularly to paragraphs starting L121, L143, L171, L266, L279 and L320, but note that other sections of the manuscript have also been modified to account for these changes. Importantly, our original interpretation remains robust – i.e. that lowland plants most likely operate by accelerating soil carbon outputs, not decelerating soil carbon inputs – but we are careful to present our conclusions with an appropriate level of caution.

– For the glasshouse experiment, I agree that the results indicate that (L115); "lowland plants accelerated microbial activity by increasing the quantity of root exudates", but not that (L112): "these findings together imply that lowland plants accelerate alpine soil C loss" because stimulating microbial activity is not per se an indicator of soil C loss. It is now well-known that the activity of microbes is not only a motor for soil C losses, but also a key mechanism leading to transformation of C inputs from plants that leads to the subsequent stabilisation of C in the soil. This is actually clearly stated further down in the manuscript when interpreting the field microbial data (L190). Furthermore, there is no direct evidence that the pots with lowland plants were losing more C than those without. Therefore, results from the glasshouse experiment could be interpreted differently: a larger fast cycling pool of soil C constituted of recently photosynthetically fixed exudates associated with higher microbial activity could well be interpreted as an early indicator of more C stabilisation, particularly since the absorbance index seems to indicate more microbially derived product in the DOC. It would have been great to measure microbial biomass C over time (as well as CUE, and mass specific growth and respiration), to see if higher respiratory activity was associated with higher biomass. The lack of differences in microbial biomass between the plant community treatments at the end of the 6 weeks does not show that the quantity of microbial biomass produced over the whole incubation period remained constant. In a word, more respiration of a larger fast cycling pool is not an indicator of future soil C loss (in the presence of plants).

We thank the Reviewer for raising this important point. On reflection, we agree that the previous version of the manuscript did not give sufficient consideration to the possibility for increased microbial activity (and, indeed, respiration) in the glasshouse experiment to signal soil carbon accumulation *via* increased microbial growth. Having said this, all pots began with the same soil and microbial biomass remained unchanged between alpine and lowland plant treatments at the end of the six-week experiment. By extension, no net microbial growth occurred during this timeframe, making it unlikely that the accelerated respiration observed under lowland plants was indicative of soil carbon accumulation. Sadly, while we can deduce that intrinsic rates of respiration were higher, we can only speculate that growth remained unchanged (no new measurements can be done since growth measurements require fresh soil). We have rewritten the respective section in the manuscript in light of this and the Reviewer’s other comments, which includes the following caveat:

L181: “These findings support the hypothesis that lowland plants have the capacity to increase soil carbon outputs relative to alpine plants by stimulating soil microbial respiration and associated CO_2_ release. While accelerated microbial respiration can alternatively be a signal of soil carbon accumulation *via* greater microbial growth^30^, such a mechanism is unlikely to have been responsible here because it would have led to an increase in microbial biomass carbon under lowland plants, which we did not observe.”

– The interpretation of the microbial variables measured in the field line up better with current conceptualisations of the role of microbes in C cycling (but overall interpretation still lacks consideration for plant C inputs). However, interpreting those data measured once 1 year after the transplant to explain the changes that happened gradually over this whole year is a risky and difficult exercise. How do we know that CUE, Rmass, Gmass etc… measured then represent what they were a day, a week, a month before? There is an attempt to deal with this timing issue by comparison with the glasshouse experiment, but only Cmic and Rmass can really be compared and it only very partially fills in the gap in time. Besides, the interpretation of this comparison can be questioned: in the glasshouse, Rmass was higher for the lowland plant pots (as compared to alpine plant at constant temperature) but actually remained constant between the comparable treatments W and WL in the field (Figure 2m). The results from the field, therefore, do not "support observations from the glasshouse experiment" in this context (L197) and neither do they "confirm (…) that this persists for at least one season" (L199). Finally, the thinking around the pulsed nature of C losses seems misplaced because there are no evidence that soil C losses had stopped after a year in the field (no measurements of soil C content are presented after that year).

With regards to plant carbon inputs, we refer the Reviewer to their previous comment for corresponding revisions. With regards to specific comparisons between the glasshouse and field experiments, we have now deleted the sentences in question and have interpreted our results as follows:

L329: “Thus, despite lower rates of ecosystem respiration overall, alpine soil microbes still respired intrinsically faster in warmed plots containing lowland plants. Moreover, accelerated microbial respiration, but not growth, implies that alpine soils had a higher capacity to lose carbon under warming, but not to gain carbon *via* accumulation into microbial biomass, when lowland plants were present^29,30^. These findings align with observations from the glasshouse experiment that lowland plants generally accelerated intrinsic rates of microbial respiration (Figure 3), although in field conditions this effect occurred in tandem with warming.”

With regards to soil carbon loss being pulsed, while there is support for such a mechanism (e.g. see manuscript references 37-40), we agree that this is one of several hypotheses and with only two timepoints we were too confident about it in the original submission. We have now reshaped this section of the manuscript entirely to be more cautious about the temporal dynamics involved (L265 onwards). For instance, the section title now reads “Lowland plant-induced soil carbon loss is temporally dynamic”. Some other notable changes are:

L286: “Importantly, lowland plants had no significant bearing over net ecosystem exchange (Figure 5a), implying that although lowland plants were associated with soil carbon loss from warmed alpine plots (Figure 1), this must have occurred prior to carbon dioxide measurements being taken and was no longer actively occurring.”

L293: “By contrast, ecosystem respiration in warmed alpine plots was depressed in the presence versus absence of lowland plants (Figure 5c). These findings generally support the hypothesis that lowland plants affect the alpine soil system by changing carbon outputs. However, they contrast with expectations that lowland plants perpetually increase carbon outputs from the ecosystem and thus raise questions about how soil carbon was lost from warmed plots containing lowland plants (Figure 1).”

L320: “Carbon cycle processes are constrained by multiple feedbacks within the soil system, such as substrate availability^36,37^ and microbial acclimation^38–40^, that over time can slow, or even arrest, soil carbon loss^37,39,40^. We thus interrogated the state of the soil system in the field experiments in the western Alps experiment to explore whether such a feedback may be operating here, in particular to limit ecosystem respiration once soil carbon content had decreased in warmed alpine plots containing lowland plants.”

L354: “Taken together, one interpretation of our findings is that the establishment of lowland plants in warming alpine ecosystems accelerates intrinsic rates of microbial respiration (Figure 3, Figure 6a), leading to soil carbon release at baseline levels of microbial biomass (Figure 1, Figure 3c), a coupled decline in microbial biomass (Figure 6c) and a cessation of further carbon loss from the ecosystem (Figure 5a, Figure 6d).”

L358: “Although such a mechanism has been reported in other ecosystems^37,40^, applying it here is speculative without additional timepoints because field soil measurements came from a single sampling event after soil carbon had already been lost from the ecosystem. For instance, an alternative mechanism could be that soil microbes acclimate to the presence of lowland plants and this decelerated microbial processes over time^38–40^.”

L368: “Beyond the mechanism for lowland plant effects on alpine soil carbon loss, it is conceivable that soil carbon loss is not isolated to a single season, but will reoccur in the future even without further warming or lowland plant arrival. This is especially true in the western Alps experiment where warming yielded a net output of carbon dioxide from the ecosystem (Figure 5a). Moreover, in our field experiments we simulated a single event of lowland plant establishment and at relatively low abundance in the community (mean ± SE relative cover: 4.7% ± 0.7%), raising the possibility that increases in lowland plant cover or repeated establishment events in the future could facilitate further decreases in alpine soil carbon content under warming.”

In my view, it is critical that the authors address the bulk density issue. Ideally, bulk density data have been collected and the soil C content results (Figure 1) can be converted to C stocks. If bulk density data have not been collected, the effect of lowland plant migration at similar densities than in the transplant experiment on bulk density should be tested and confirmed to be neutral.That being done, dC/dt = inputs – decomposition.

We refer the Reviewer to our response to their previous comment, which we hope has resolved this issue.

Nearly every aspect of the dataset should be re-interpreted in terms of a shifting balance between inputs and decomposition, rather than simply in terms of more or less microbial respiration. It could mean having to conduct additional analyses on the plant traits. To put it bluntly, it feels like the microbial data were interpreted as if collected from a soil incubation experiment with no plants (this is not totally fair of course because plant traits were measured, but not really interpreted in a way that is helpful to conclude on the input/decomposition balance).

We refer the Reviewer to our response to their previous comments, which we hope has resolved this issue.

In my view, the manuscript needs nearly a complete re-write, but I reiterate that this is a novel, valuable and properly (but perhaps not completely) analysed (statistically) data set.

We thank the Reviewer again for this positive comment. We have completely overhauled and re-written the manuscript in response to these and the other reviewer comments and are confident that all issues have been appropriately addressed.

Reviewer #4:This manuscript took alpine grasslands as a model system and investigated whether lowland herbaceous plants contributed to the short-term dynamics of soil carbon under the context of climate warming. The authors find that warming individually does not render significant changes in alpine soil carbon, but corporately causes ~52% of carbon loss with lowland herbaceous plants in two short periods of field experiments. They further show that alpine soil carbon loss is likely mediated by lowland herbaceous plants through root exudation, soil microbial respiration, and CO2 release. This work adds in an interesting way to the ongoing debate on whether a positive climate feedback will be mediated by plant uphill range expansion in alpine grasslands, where climate warming may lead to a rapid loss of soil carbon.The claims of this manuscript are well supported, but some aspects of background information in the studied alpine systems and field experiment design need to be clarified.1) There is an extremely high level of carbon stored in the alpine soils (Figure 1). Climate warming will certainly lead to a great loss of soil carbon in the study systems that could contribute to the positive climate feedback. However, it is unclear for me how the effects of climate warming on soil carbon are relevant to the ongoing climate change in the studied alpine grasslands. It is therefore reasonable to provide more background information about ongoing climate change, and whether the simulated climate warming (i.e., 2.8 oC in central alps and 5.3 oC in western alps, Line 328-329) is realized as real-world climate change in the local systems. In addition, it seems that the manuscript aims to address a question that is of global concern, but my concern is about how the findings could be generalized to other regions.

We thank the Reviewer for pointing this out. With regards to the amount of soil carbon stored in the alpine soils, we refer the Reviewer to comments from Reviewer #2. With regards to the magnitude of warming expected in mountain regions, we agree with the Reviewer that the original submission lacked context. We have therefore added specific values as suggested:

L59: “They are experiencing both rapid temperature change (0.4 to 0.6 ºC per decade)^18,19^ and rapid species immigration^20^…”

With regards to how findings could be generalised to other regions or ecosystems, this is an important point that requires further research – and which we raise in the concluding paragraph. However, we see that we could have been more explicit about validating our findings in other mountain regions, so we have amended the sentence in question as follows:

L400: “Future work should focus on testing the conditions under which this feedback could occur in different mountain regions, as well as other ecosystems, experiencing influxes of range expanding plant species, on quantifying how deeply it occurs in shallow alpine soils, and on estimating the magnitude of the climate feedback given both ongoing warming and variation in rates of species range shifts.”

2) I understand that the manuscript considers elevation as a natural gradient of climate change, which makes it possible to compare soil carbon dynamics in lowlands with alpine grasslands under climate warming. I also understand that the authors have done everything they can to control for the disturbances caused by transplanting that has been well justified by the supplementary data (e.g., Figure S6). However, it is unclear how the authors controlled for the influences of other factors given there are huge differences between lowlands and alpine grasslands, such as differences in wind, solar radiation, humidity, and the length of growing season.

This is an excellent point. We note that Reviewer #1 also raised this point, so we refer the Reviewer to our response to that comment for further details.

3) It is generally known that different species respond to climate warming differently. Some species may be sensitive to climate warming and have traits aiding to dispersion that could expand their living ranges to some degree, while others may adjust themselves to adapt to climate warming and may not migrate to alpine systems. It is therefore cautious to assume that all the lowland species have the same dispersal ability. In other words, it is unclear how lowland plant species are selected for the field transplanting experiment (Line 284-290). Do all the lowland plant species selected have the potential to migrate to alpine systems?

This is an excellent question. In short, the specific dispersal abilities of lowland species used are currently unknown and will certainly vary. However, all are widespread and we assume have the capacity to migrate to higher elevations, given that horizontal distances between high and low elevation sites were in both cases less than 2 km. We now clarify this in the manuscript as follows:

L433: “While exact dispersal distances for selected lowland species are unknown, all species are widespread and are expected to migrate uphill under warming^8^ and the horizontal distance between high and low sites in the field experiments was always less than 2 km.”

4) The authors acknowledge that "we did not perform a reverse transplantation (that is, from low to high elevation), so we cannot entirely rule out the possibility that transplantation of any community to any new environment could yield a loss of soil carbon" (Line 318-320). When I read the title "lowland plant migrations into alpine grasslands …", I thought lowland plant species that were transplanted from low to high elevation. In fact, it is just the opposite to my thoughts. Without performing a reverse transplantation experiment, I am not sure the conclusion will stand that "lowland plant migrations into alpine grasslands amplify soil carbon loss under climate warming". In addition, it is unclear whether lowland plant effects stand alone or depend on climate warming based on the results in Figure 1 that lowland plant treatment is missing, and it is impossible to test the interactions between lowland plant and climate warming.

We apologise for the confusion. This comment echoes other comments from Reviewer #1 asking us to be more explicit about the treatments used when interpreting findings, to caveat the step in logic from transplantation to warming and to acknowledge throughout the manuscript that lowland plant effects were dependent on transplantation in the field experiment.

We therefore refer the Reviewer to our responses to those comments for details on how we resolved this. We have also modified the title and abstract to more accurately represent the experimental design, as follows:

Title: “Lowland plant arrival in alpine ecosystems facilitates soil carbon loss under experimental climate warming”

L30: “Here we used two whole-community transplant experiments and a follow-up glasshouse experiment to determine whether the establishment of herbaceous lowland plants in alpine ecosystems influences soil carbon content under warming. We found that warming (transplantation to low elevation) led to a negligible decrease in alpine soil carbon content, but its effects became significant and 52% ± 31% (mean ± 95% CIs) larger after lowland plants were introduced at low density into the ecosystem.”

With regards to testing the interaction between warming and lowland plants, while we acknowledge that not performing a fully-factorial design limited our ability to explicitly separate lowland plant versus warming effects on alpine soil, both are occurring simultaneously due to climate warming and we thus focussed effort on simulating such a scenario with greater experimental replication and at multiple locations. We note that Reviewers #1, #2 and #3 thought that this approach was robust. Importantly, the statistical analyses performed are valid for such an experimental design, and we have clarified and nuanced our interpretation throughout to avoid reaching beyond it.